# PPP2R1A regulates migration persistence through the NHSL1-containing WAVE Shell Complex

Yanan Wang[1], Giovanni Chiappetta[2,7], Raphaël Guérois [3,7], Yijun Liu[4,7], Stéphane Romero [1,7], Daniel J. Boesch[4], Matthias Krause [5], Claire A. Dessalles[6], Avin Babataheri[6], Abdul I. Barakat [6], Baoyu Chen [4], Joelle Vinh [2], Anna Polesskaya [1] ✉ & Alexis M. Gautreau [1] ✉

The RAC1-WAVE-Arp2/3 signaling pathway generates branched actin networks that power lamellipodium protrusion of migrating cells. Feedback is thought to control protrusion lifetime and migration persistence, but its molecular circuitry remains elusive. Here, we identify PPP2R1A by proteomics as a protein differentially associated with the WAVE complex subunit ABI1 when RAC1 is activated and downstream generation of branched actin is blocked. PPP2R1A is found to associate at the lamellipodial edge with an alternative form of WAVE complex, the WAVE Shell Complex, that contains NHSL1 instead of the Arp2/3 activating subunit WAVE, as in the canonical WAVE Regulatory Complex. PPP2R1A is required for persistence in random and directed migration assays and for RAC1-dependent actin polymerization in cell extracts. PPP2R1A requirement is abolished by NHSL1 depletion. PPP2R1A mutations found in tumors impair WAVE Shell Complex binding and migration regulation, suggesting that the coupling of PPP2R1A to the WAVE Shell Complex is essential to its function.

Cell migration is a critical process for animal cells, especially in the embryo, but also in the adult. For example, immune cells constantly patrol the organism to fight infections. Nucleation of branched actin by the Arp2/3 complex fuels membrane protrusions called lamellipodia. This type of cell migration is characterized by its persistence that can be seen even in random migration and measured by the time during which direction is maintained once it is established[1]. Arp2/3-mediated persistence also favors directed migration towards higher concentrations of extracellular matrix proteins in a process called haptotaxis[2].

The WAVE proteins activate Arp2/3 at the cell cortex and in membrane protrusions[3]. WAVE proteins are embedded in a stable multiprotein complex containing four other subunits, CYFIP, NCKAP, ABI, and BRK1. This stable complex has been purified from various sources and its exact composition differ depending on the paralogous subunits expressed[4,5]. Most importantly, native and reconstituted recombinant WAVE complexes were shown to keep inactive the otherwise constitutively active WAVE protein[6,7]. Therefore, this stable complex is commonly referred to as the WAVE regulatory Complex (WRC). GTP-bound RAC1 activates the WRC through an allosteric mechanism[8,9]. The WCA domain of WAVE that binds and activates Arp2/3 is masked within the WRC, but becomes exposed upon RAC1-mediated activation[10–12]. The RAC1-WAVE-Arp2/3 pathway is critical for

[1]Laboratory of Structural Biology of the Cell (BIOC), CNRS UMR7654, École Polytechnique, Institut Polytechnique de Paris, 91120 Palaiseau, France. [2]Biological Mass Spectrometry and Proteomics (SMBP), ESPCI Paris, Université PSL, LPC CNRS UMR8249, 75005 Paris, France. [3]Université Paris-Saclay, CEA, CNRS, Institute for Integrative Biology of the Cell (I2BC), 91198 Gif-sur-Yvette, France. [4]Roy J. Carver Department of Biochemistry, Biophysics and Molecular Biology, Iowa State University, Ames, IA 50011, USA. [5]Randall Centre for Cell and Molecular Biophysics, King's College London, New Hunt's House, Guy's Campus, London SE1 1UL, UK. [6]LadHyX, École Polytechnique, Institut Polytechnique de Paris, 91120 Palaiseau, France. [7]These authors contributed equally: Giovanni Chiappetta, Raphaël Guérois, Yijun Liu, Stéphane Romero. ✉e-mail: anna.polesskaya@polytechnique.edu; alexis.gautreau@polytechnique.edu

development and normal adult life but is also involved in cancer[13–15]. Genes encoding subunits of the WAVE and Arp2/3 complexes are overexpressed in a variety of cancers, and this overexpression is associated with high tumor grade and poor patient prognosis[16].

Cell migration is finely regulated at all molecular levels. Each positive component required to generate cortical branched actin, RAC1, WRC, and Arp2/3, appears to be counteracted by inhibitory proteins, CYRI[17,18], NHSL1[19], and ARPIN[20,21], respectively. NHSL1 belongs to the family of Nance-Horan Syndrome (NHS) proteins, which contain an N-terminal WAVE Homology Domain (WHD), as in WAVE proteins[22]. The WHD is the main structural domain that embeds WAVE proteins in the WRC[8,11], raising the possibility that WRC might contain NHS family proteins instead of WAVE proteins at some point in their life cycle. This intriguing possibility, however, has never been reported to date. Instead NHSL1 has been shown to interact with the WRC through the C-terminal SH3 domain of the WRC subunit ABI1[19].

Regulatory circuits of cell migration involve feedback and feed-forward loops[23,24]. ARPIN was shown to inhibit Arp2/3 only when RAC1 signaling was on, rendering lamellipodia unstable once they were formed instead of preventing their formation[20]. Positive feedback that sustains membrane protrusion at the front is thought to be responsible for the persistence of cell migration. Indeed, signaling pathways are not linear, and actin polymerization activate WRC further in space and time in propagating waves[25,26] and promotes WRC turn-over[27]. Biochemical signaling in feedback is constrained by cell mechanics and in particular membrane tension, which appears as a central component to sustain lamellipodial protrusion at the leading edge[28,29]. Feedback signaling is so challenging to dissect that mathematical modeling, computational simulations and machine learning are often required to interpret complex observations that would not fit into simple and linear models[30,31].

Here we used criteria of feedback signaling to uncover a molecular machine that regulates migration persistence. Using differential proteomics, we identified the PPP2R1A protein as a migration regulator. PPP2R1A is a well-characterized scaffold subunit of the PP2A complex, an abundant and heterogeneous phosphatase that regulates several signaling pathways[32]. PP2A phosphatase activity has previously been connected to cell migration and tumor cell invasion[33–35], but we found no evidence that the precise role of PPP2R1A in regulating migration persistence depends on PP2A phosphatase activity. The control of migration persistence by PPP2R1A rather depended on its interaction with the NHSL1-containing WAVE Shell Complex (WSC). PPP2R1A is mutated in various cancers, particularly gynecologic cancers[36], and frequent tumor-associated mutations of PPP2R1A were found impair its interaction with the WSC.

## Results

### Identification of PPP2R1A as a regulator of migration persistence

To identify partners of WRC, we used proteomics after tandem affinity purification (TAP) of the ABI1 subunit. We reasoned that partners differentially associated with ABI1 when RAC1 is activated and when Arp2/3 is inhibited would be good candidates to regulate migration persistence and feedback signaling. As a model system, we used the human mammary epithelial cell line MCF10A, which is immortalized but not transformed. We thus isolated stable cell lines expressing FLAG-GFP ABI1 from both parental cells and a genome-edited derivative, where the Q61L mutation affects one allele of *RAC1* and renders the RAC1 GTPase deficient and hence constitutively active (Supplementary Fig.1). MCF10A cells expressing RAC1 Q61L showed more persistent migration than parental cells[37]. We treated cells with the Arp2/3 inhibitory compound CK-666 to modulate the binding of potential WRC partners involved in signaling feedback from branched actin. The CK-666 treatment did not impact RAC1 activation levels compared to controls (Supplementary Fig.1).

After the two successive FLAG and GFP immunoprecipitations (Fig. 1a), we identified by mass spectrometry proteins specifically associated with ABI1, but not with FLAG-GFP control (Supplementary Table 1). We performed label-free quantification of 3 biological replicates and calculated in all four conditions the relative abundance of associated proteins after normalization by the amount of the ABI1 bait protein retrieved in each condition (see Supplementary Methods). The WRC subunits did not vary considerably in the different conditions (Fig.1b). Among the hits, we recognized several described functional partners of the WRC such as the Nance-Horan Syndrome family proteins NHS and NHSL1[19,22], IRSp53, encoded by *BAIAP2*[38], and lamellipodin, encoded by *RAPH1*, which was 3-fold more associated with WRC when RAC1 was activated, as previously reported[39].

Our attention was drawn to PPP2R1A, a scaffold subunit of the PP2A phosphatase complex, which showed variations in the two-fold range when RAC1 was activated and when CK-666 was applied to cells. The amount of PPP2R1A associated with ABI1 decreased when RAC1 was constitutively activated (Fig.1b). Surprisingly, CK-666 induced opposite PPP2R1A variations in parental and RAC1 Q61L cells. Catalytic subunits of the same PP2A phosphatase complex, encoded by the two paralogous genes, *PPP2CA* and *PPP2CB*, were also detected in the list of ABI1 partners, but interestingly, these subunits did not show the same variations as PPP2R1A when CK-666 was applied to cells.

We depleted PPP2R1A from MCF10A cells using siRNA pools and tested cells in a 2D random migration assay. Lamellipodia appeared less developed in PPP2R1A-depleted cells, which, as a result, were less spread than control cells (Supplementary Movie 1). PPP2R1A-depleted cells did not maintain the direction of migration over time as well as controls (Fig. 1c). We analyzed RAC1 activation and levels in control and PPP2R1A-depleted cells (Supplementary Fig. 1). PPP2R1A depletion was associated with a tendency to decrease RAC1 activity and a slight, but significant, down-regulation of RAC1 levels. Strikingly, the decrease in migration persistence upon PPP2R1A depletion was not significant when the same assay was performed using MCF10A cells expressing RAC1 Q61L (Fig. 1d and Supplementary Movie 2). This was surprising given that migration persistence is increased in MCF10A RAC1 Q61L[37], but was in agreement with our observation that PPP2R1A interacted less with ABI1 when RAC1 is constitutively active. We then isolated a stable MCF10A cell line that expresses FLAG-GFP PPP2R1A. The overexpression of tagged PPP2R1A was moderate, estimated by densitometry of PPP2R1A Western blots to be in the order of two-fold (1.96 ± 0.32, mean ± sem of three triplicates, $P < 0.05$, $t$-test), with a concomitant decrease in endogenous PPP2R1A (Fig.1e). PPP2R1A overexpression had no effect on levels of RAC1, WAVE2, or WRC (Supplementary Fig. 1 and Supplementary Fig.2), but was sufficient to increase migration persistence in the assay (Fig. 1e and Supplementary Movie 3). The loss- and gain-of-function experiments in MCF10A cells thus indicate that migration persistence depends on PPP2R1A levels.

To validate our findings in another cell system, we used the invasive breast cancer cell line MDA-MB-231 that we embedded in 3D collagen gels. PPP2R1A depletion dramatically decreased migration persistence in this setting as well (Fig. 1f and Supplementary Movie 4). Overexpression of PPP2R1A, however, did not affect migration persistence in either MDA-MB-231 cells, or in MCF10A cells expressing RAC1 Q61L (Supplementary Fig. 2). Migration persistence of cells expressing constitutively active RAC1 Q61L thus appears independent of PPP2R1A levels, unlike parental MCF10A. Other parameters of cell migration necessary for a complete description of cell trajectories, such as cell speed and Mean Square Displacement (MSD), were also altered in some of these genetic perturbations, but inconsistently (Supplementary Fig. 2), as previously reported[20,37], confirming that migration persistence, but not these other parameters, appears as the primary target of the RAC1-WAVE-Arp2/3 pathway[23].

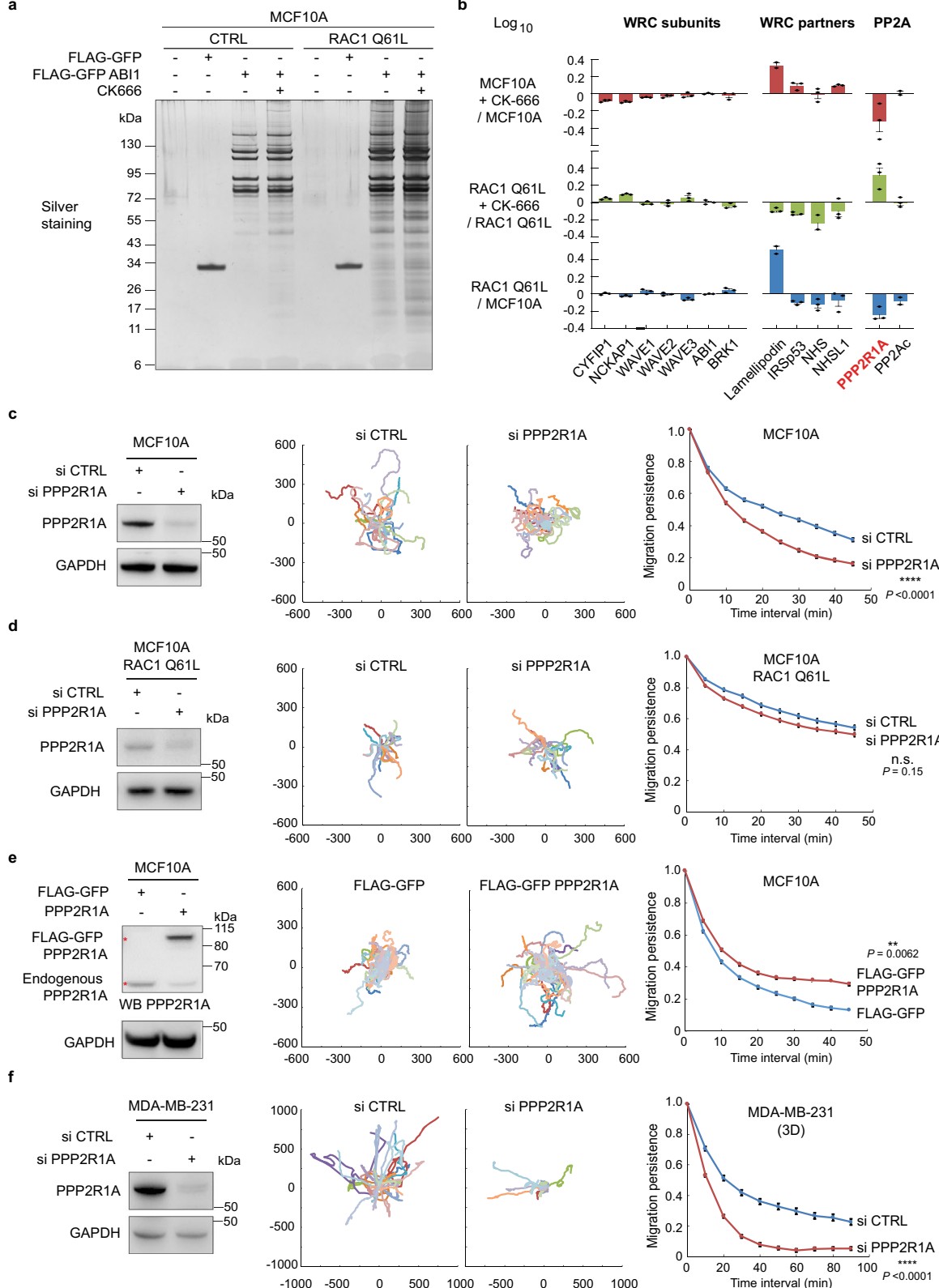

## PPP2R1A interacts with the NHSL1-containing WAVE Shell Complex

To decipher how PPP2R1A regulated migration of MCF10A and MDA-MB-231 cells, we characterized its partners by proteomics. We purified FLAG-GFP PPP2R1A by TAP (Fig. 2a) and identified its partners by mass spectrometry (Supplementary Table 2). As expected, we detected in MCF10A and MDA-MB-231 cells the catalytic phosphatase subunit and numerous regulatory subunits of PP2A complexes (PPP2R1B, PPP2R2D, PPP2R3B, PPP2R5A, PPP2R5B, PPP2R5C, PPP2R5D, and PPP2R5E) and its larger STRIPAK derivatives (STRN, STRIP2, STRN4, MOB4). PPP2R3C was detected associated with PPP2R1A in MCF10A, but not MDA-MB-231 cells. We expected to find the whole WRC in line with the TAP purification of ABI1. However, we detected only 4 of the 5 subunits of the WRC (Fig. 2b), namely CYFIP1, NCKAP1, ABI1, and BRK1. BRK1 was

**Fig. 1 | Identification of PPP2R1A as an ABI1 partner that regulates migration persistence. a** MCF10A parental cells or genome-edited MCF10A cells where a *RAC1* allele encodes the active RAC1 Q61L mutant form were stably transfected with plasmids expressing FLAG-GFP or FLAG-GFP ABI1. Cells were treated or not with 100 μM CK666 for 16 h in order to block the polymerization of branched actin and thus the suspected feedback loop that mediates migration persistence. FLAG-GFP proteins were then purified by Tandem Affinity Purification and associated proteins were revealed by silver staining of SDS–PAGE gels. **b** Label-free quantification of selected proteins identified by mass spectrometry to be associated with ABI1. 3 biological repeats, mean ± sem of the $\log_{10}$ of fold changes are plotted. **c** MCF10A cells were transfected with pools of control (CTRL) or PPP2R1A siRNAs and analyzed by Western blots with PPP2R1A and GAPDH antibodies. Cell trajectories and

migration persistence extracted from 2D migration of single MCF10A cells transfected with indicated siRNAs. Tracking 7.5 h, $n = 17$ cells. **d** Same experiment as in **c** with MCF10A RAC1 Q61L cells. Tracking 6 h, $n = 18$ cells. **e** MCF10A cells were stably transfected with plasmids expressing FLAG-GFP or FLAG-GFP PPP2R1A. Migration was analyzed as in **c**. Tracking 10 h, $n = 37$ cells. **f** Same experiment as in **c** with MDA-MB-231 cells analyzed in 3D collagen type I gels. Tracking 12 h, $n = 25$ cells. Most cell trajectories upon PPP2R1A depletion are so short that they cannot be well distinguished, because they overlap in the center of the graph. Trajectory plots are labeled in μm. Three biological repeats of each experiment yielded similar results, only one is displayed. Data are shown as mean ± SEM. Statistical significance was calculated with custom-made R programs and *P* values are indicated. \*\**P* < 0.01; \*\*\*\**P* < 0.0001; n.s. not significant. Source data are provided as a Source Data file.

only detected in MCF10A and not in MDA-MB-231 cells, but this is likely because BRK1 generates only a few tryptic peptides from its short sequence of 75 amino-acids. Surprisingly, the 3 WAVE proteins were absent from the TAP purification from both cell lines. We searched for NHS family proteins, which were reported to contain an N-terminal WAVE homology domain (WHD) similar to that responsible for WAVE incorporation into WRC[8,22], and found NHSL1 in the TAP purification from both cell lines. We validated the presence of NHSL1 and all WRC subunits except WAVE by western blots (Fig. 2c). We never detected any WAVE protein in TAP purification of PPP2R1A. When NHSL1 was depleted from cells using siRNAs, PPP2R1A immunoprecipitation did not retrieve WRC subunits, but still retrieved the PP2A catalytic subunit (Fig. 2d). This result suggests that NHSL1 is the subunit of this alternative complex that is recognized by PPP2R1A.

To examine the composition of the alternative complex devoid of WAVE, we performed TAP to select for the presence of both PPP2R1A and the WRC subunit BRK1. We performed sequential FLAG - PC immunoprecipitations from the stable MCF10A cell line expressing both FLAG-GFP PPP2R1A and PC-HA BRK1. We indeed retrieved the WRC without WAVE but containing NHSL1 instead (Fig. 2e and Supplementary Table 3). This TAP purification of both PPP2R1A and BRK1 did not completely exclude PP2A subunits, since it also contained a subset of the subunits found in TAP of PPP2R1A alone, namely PPP2CA, PPP2R2A, PPP2R5D, PPP2R5E, and STRN. This result suggests that PPP2R1A can bind to this alternative WRC while being part of PP2A complexes. To our knowledge, a physiological form of WRC without a WAVE subunit and containing an NHS family protein instead, has never been previously reported. NHSL1 was recently reported to interact with the complete canonical WRC through the ABI1 SH3 domain[19]. PPP2R1A thus does not bind to all pools of NHSL1 molecules, but appears to select a specific conformation of NHSL1, where NHSL1 fully replaces WAVE within its complex. We call this alternative to WRC, the NHSL1-containing WAVE Shell Complex (WSC), to emphasize that this multiprotein complex, which normally embeds the WAVE subunit and regulates its activity, has lost its Arp2/3-activating WAVE subunit.

To obtain a structural model of the WSC, we used the AlphaFold2 program[40–42]. We were able to generate a high-confidence structural model (see Supplementary Methods) where NHSL1 interacted with the CYFIP1, NCKAP1, BRK1, and ABI2 subunits (Fig. 2f). Core to this interaction is the WAVE homology domain (WHD, residues 30–90) that forms a triple coiled coil with BRK1 and ABI2 (Supplementary Fig. 3), as the N-terminal helix of the WAVE1 subunit in the case of the WRC[8,43]. Unlike WAVE1 in the WRC, no other region of NHSL1, beyond the proline 97, is predicted to interact with WSC subunits. NHSL1 interactions within WSC appear less extensive than WAVE interactions within WRC.

Using in vitro translation with $^{35}$S-methionine, we were able to reconstitute the WSC, as we had previously reconstituted the WRC[5]. The N-terminal fragment of NHSL1, containing the WHD, retrieves CYFIP1, NCKAP1, ABI1, and BRK1, but excludes WAVE2 (Fig. 3a). In contrast, the C-terminal fragment of NHSL1, containing ABI1 SH3 binding sites, can interact with the WRC containing WAVE2, as previously reported[19]. Competition experiments using simultaneous

translation of WAVE and NHSL1 WHD were consistent with structural predictions, since WAVE2 can displace NHSL1 WHD from the 'shell' composed of CYFIP1, NCKAP1, ABI1, and BRK1 (Fig. 3b). The NHSL1 WHD, however, is unable to displace WAVE2, in line with the AlphaFold2 structural model where NHSL1 exhibited a smaller interaction surface within the WSC than WAVE within the WRC. We then used a method designed to purify recombinant WRC, based on the expression of various subunits in bacteria and insect cells to purify the WSC[44]. It was indeed sufficient to replace WAVE1 N-terminus by a fragment of NHSL1 containing its WHD to purify a WSC (Fig. 3c) that behaved like the WRC in gel filtration (Fig. 3d). Direct observation of reconstituted WSC and WRC by electron microscopy after negative staining showed that the two complexes had similar sizes and shapes (Fig. 3e).

To obtain an estimate of the amount of WSC in the cell, we analyzed exponentially growing MCF10A cells, because we found that starvation down-regulates WSC (Supplementary Fig. 4). The distribution of WSC and WRC subunits were analyzed by ultracentrifugation along sucrose gradients. PPP2R1A overlapped with the catalytic phosphatase subunit of the PP2A complex (Supplementary Fig. 5). NHSL1 was in between WRC and PP2A complexes. We obtained *PPP2R1A* and *NHSL1* KO MCF10A cell lines using CRISPR/Cas9 (Supplementary Fig. 5). Importantly, the distribution of none of these subunits was altered upon depletion of PPP2R1A or of NHSL1 (Supplementary Fig. 5), suggesting that the WSC is not an abundant complex. In an attempt to evaluate the relative amount of WSC compared to that of WRC, we performed sequential immunoprecipitations to distinguish WSC from WRC after a first selection of BRK1-containing complexes. We estimated that the WSC is 7 to 8 times less abundant than the WRC (Supplementary Fig. 4). We also examined phosphosites when PPP2R1A was depleted or not, but found no differentially phosphorylated site on the WSC, on the WRC or on RAC1 itself (Supplementary Fig. 6 and Supplementary Table 4).

We then sought to examine PPP2R1A localization and used for this purpose B16-F1 cells that develop prominent lamellipodia. Cells were transfected with GFP-NHSL1 and mScarlet-PPP2R1A. PPP2R1A was colocalized with NHSL1 at the lamellipodial edge (Fig. 4a and Supplementary Movie 5). NHSL1 was strongly enriched at the lamellipodial edge, as previously described[19], whereas PPP2R1A extended far beyond, in the entire width of the lamellipodium (Fig. 4b). Arp2/3 is incorporated into branched actin networks and extends backwards through the actin retrograde flow. We thus examined the colocalization of mScarlet-PPP2R1A with the GFP-tagged Arp2/3 subunit ARPC1B. Both proteins colocalized in the width of lamellipodia (Fig. 4c and Supplementary Movie 6), but ARPC1B distribution declined steeply away from the edge, whereas PPP2R1A did not exhibit such a gradient (Fig. 4d). These analyses suggest that PPP2R1A is a lamellipodial protein that is likely to interact with WSC at the cell edge.

To examine PPP2R1A dynamics, we performed Fluorescence Recovery After Photobleaching (FRAP) in the lamellipodia of B16-F1 cells. A fraction of GFP-PPP2R1A was immobile, but the major fraction in the majority of cases recovered fast with a $t_{1/2}$ of less than a second

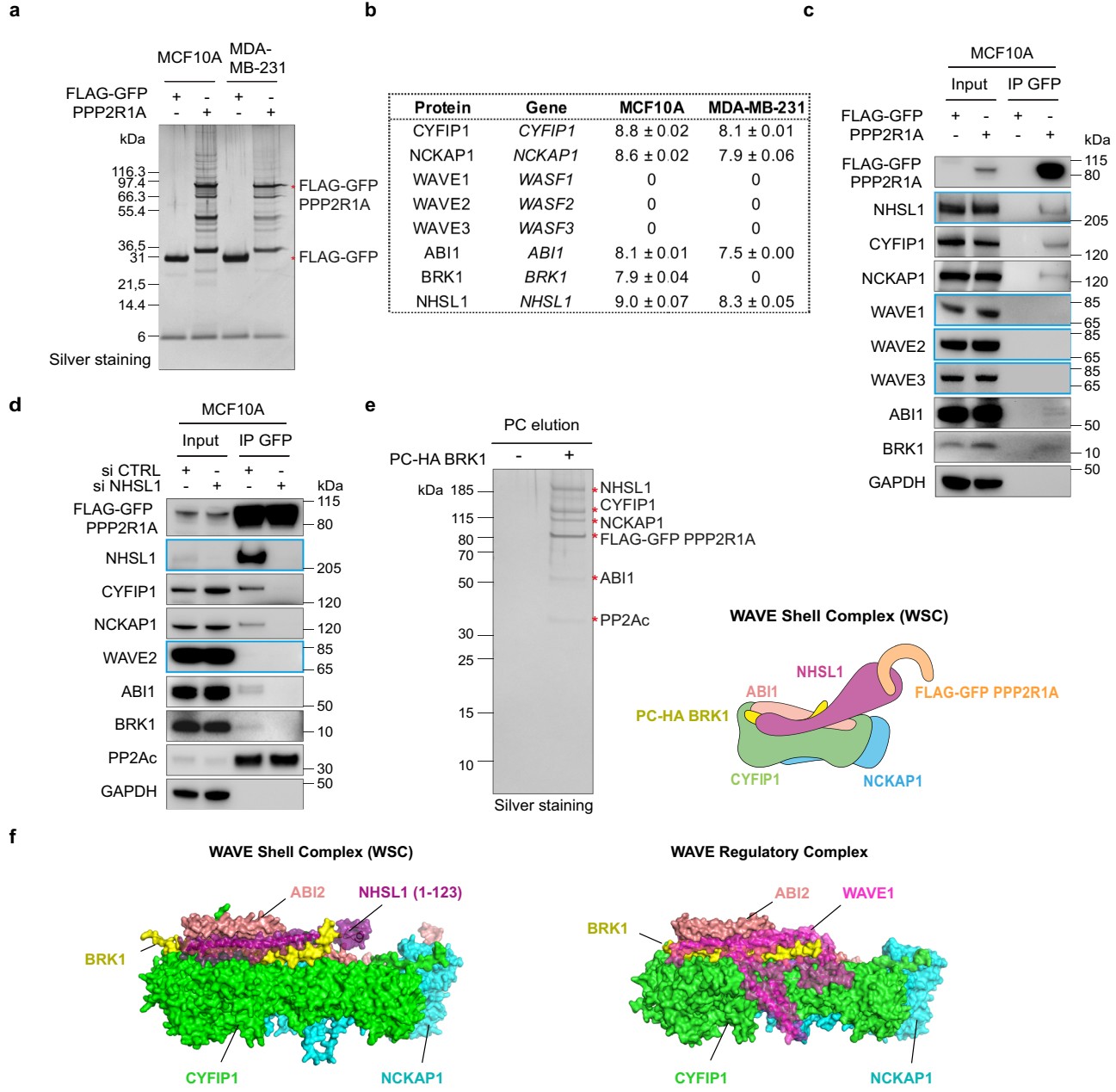

**Fig. 2 | PPP2R1A interacts with the WAVE Shell Complex (WSC) that contains NHSL1 instead of WAVE proteins. a** MCF10A or MDA-MB-231 cells stably expressing FLAG-GFP or FLAG-GFP PPP2R1A were subjected to FLAG-GFP Tandem Affinity Purification and purified proteins were resolved by SDS−PAGE and silver stained. The same samples were used for mass spectrometry and Western blots. Three biological repeats with similar results. **b** Label-free quantification (LFQ) of proteins identified by mass spectrometry. In both cell lines, NHSL1 is present at the expense of WAVE subunits. Log10 of LFQ values, mean ± sem of three biological repeats. **c** GFP immunoprecipitates were analyzed by Western blots with the indicated antibodies. Three biological repeats with similar results. **d** PPP2R1A associates with the WSC through NHSL1. MCF10A cells expressing FLAG-GFP PPP2R1A were transfected with pools of siRNAs targeting NHSL1 or non-targeting siRNAs (CTRL).

GFP immunoprecipitates were analyzed by western blots. Three biological repeats with similar results. **e** A MCF10A cell line stably transfected with two plasmids expressing FLAG-GFP PPP2R1A and PC-HA BRK1 was subjected to FLAG-PC Tandem Affinity Purification to purify the WSC. WSC composition was analyzed by mass spectrometry. Silver stained SDS−PAGE of purified WSC. The identity of WSC subunits and the lack of WAVE2 was confirmed using Western blots. 3 biological repeats with similar results. **f** Comparison of the structural model of the WSC obtained using AlphaFold2 with the X-ray crystal structure WAVE Regulatory Complex (PDB:3P8C [https://www.wwpdb.org/pdb?id=pdb_00003p8c]). The WSC was composed of NHSL1 (1-123) in purple, CYFIP1 in green, NCKAP1 in cyan, BRK1 in yellow and ABI2 (1−160) in salmon.

---

(Fig. 5a–c and Supplementary Movie 7). The recovery was not different at the lamellipodial edge or in the width of the lamellipodium. PPP2R1A recovery was homogenous throughout the lamellipodium width, unlike that of Arp2/3, which exhibits a retrograde flow[45]. When GFP-NHSL1 and mScarlet-PPP2R1A were simultaneously photobleached, their recovery time at the lamellipodial edge was not different (Fig. 5d, e and Supplementary Movie 8). These FRAP results are compatible with PPP2R1A transiently interacting with actin filaments in the width of lamellipodia and with NHSL1 at the lamellipodial edge.

## The requirement of PPP2R1A in cell migration and actin polymerization depends on the presence of NHSL1

NHSL1 was previously described as a negative regulator of cell migration[19]. It was thus surprising to find PPP2R1A and NHSL1

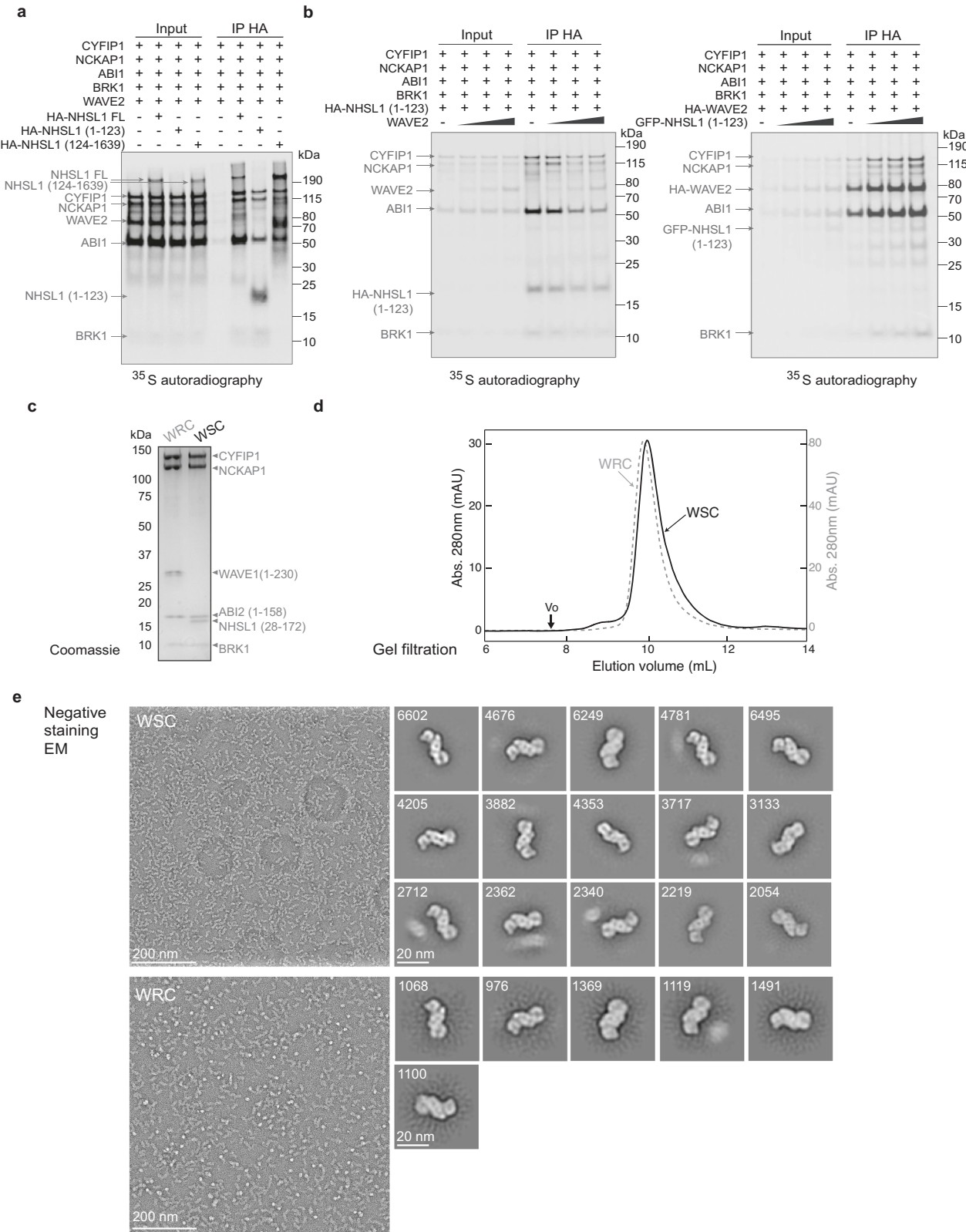

biochemically associated. We have knocked down either PPP2R1A, or NHSL1, as well as both proteins simultaneously in order to study epistasis. In line with previous findings, NHSL1 depletion strongly promoted migration persistence in random migration of single MCF10A cells. The decline of migration persistence upon PPP2R1A depletion was abolished by NHSL1 depletion (Fig. 6a and Supplementary Movie 9). Despite their opposite effects, neither of the two

migration regulators appears to take over. The simultaneous depletion of both regulators does not significantly affect migration persistence compared to controls, as if the signaling circuit containing the PPP2R1A and NHSL1 was simply dispensable in this random migration assay. We then examined directional migration in a gradient of fibronectin to further challenge cells. This haptotactic assay was previously shown to depend on Arp2/3 activity[2]. PPP2R1A-depleted cells were

**Fig. 3 | Reconstitution of the WAVE Shell Complex (WSC). a** In vitro translation of plasmids encoding subunits encoding the WRC with NHSL1 full length (FL) or indicated fragments in an untagged form or tagged with HA or GFP, as indicated. Immunoprecipitation targeting the HA tag pulls down the WSC since the HA tagged NHSL1 N-terminal fragment (1–123), containing the WHD, excludes WAVE2. In contrast, the NHSL1 C-terminal fragment (124–1639) associates with the WRC containing WAVE2. **b** Competition experiments using in vitro translation. WAVE2 displaces NHSL1 WHD for binding the shell subunits CYFIP1, NCKAP1, ABI1, and BRK1, but NHSL1 WHD does not displace WAVE2. **c** Reconstitution of recombinant WSC and WRC. Coomassie-blue stained SDS–PAGE gel, where each lane contains 3 pmol of purified complex. **d** Gel filtration chromatography of recombinant WSC and WRC using a 24 ml Superdex 200 column. Vo void volume. Recombinant complexes are pure, assembled and do not aggregate. **e** Electron microscopy of recombinant WSC and WRC after negative staining. Representative raw micrographs, 2D class averages and number of particles used to derive the class averages are shown. Note the very similar shapes of the two complexes, particularly on the first three class averages that were selected to display similar orientations. Experiments were performed once in **a** and **e** for the WSC, twice in **e** for the WRC and three times in **c** and **d** with similar results.

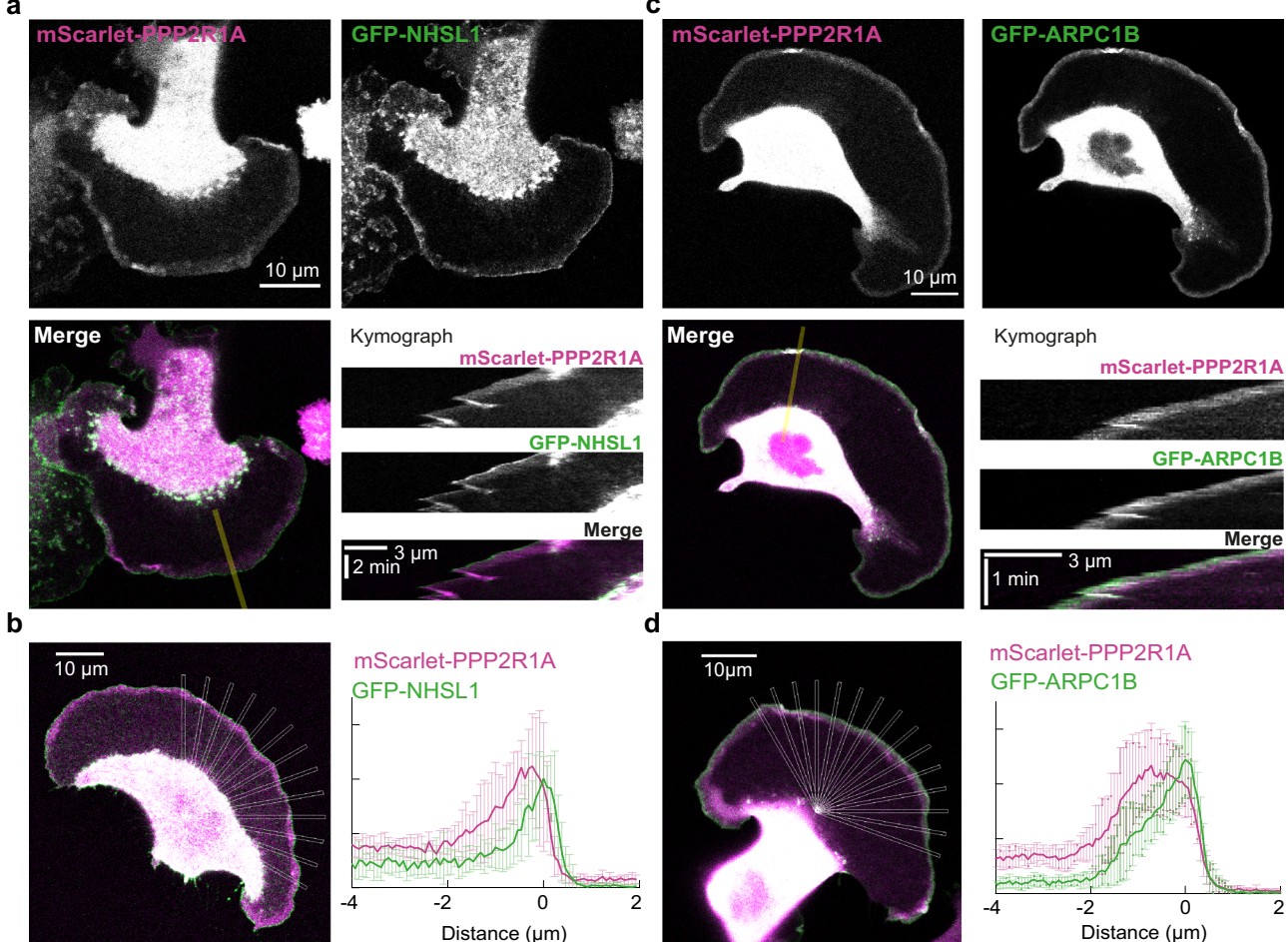

**Fig. 4 | PPP2R1A colocalizes with branched actin in the width of the lamellipodium and colocalizes with NHSL1 at the lamellipodium edge. a** B16-F1 cells were transiently transfected with mScarlet-PPP2R1A and GFP-NHSL1. A kymograph was drawn along the line shown on the merge. **b** Overlap of PPP2R1A and NHSL1 over multiple radial line scans registered to the cell edge. Data are shown as mean ± SD, n = 13 line scans. **c** B16-F1 cells were transiently transfected with mScarlet-PPP2R1A and the Arp2/3 subunit GFP-ARPC1B. A kymograph was drawn along the line shown on the merge. **d** Overlap of PPP2R1A and ARPC1B over multiple radial line scans registered to the cell edge. Data are shown as mean ± SD, n = 15 line scans. Source data are provided as a Source Data file.

strongly impaired in their ability to migrate toward higher concentrations of fibronectin (Fig. 6b and Supplementary Movie 10). In contrast, depletion of the migration inhibitor NHSL1 rendered cells slightly, but significantly, more efficient at directed migration. In this assay, as in random migration, simultaneous depletion of PPP2R1A and NHSL1 abolished the effects of each single depletion. In two different cell migration assays, PPP2R1A and NHSL1 thus depend on each other.

To establish that the effects of the PPP2R1A-NHSL1 signaling circuit were due to actin polymerization, we set up an in vitro reconstitution assay of actin polymerization in extracts prepared from MCF10A cells. Actin polymerization at the surface of glutathione beads was monitored by the incorporation of fluorescent actin. GTP-bound RAC1 Q61L was able to trigger the polymerization of actin structures

from the surface of the beads (Fig. 7a). These structures were discrete fibers, not the dense meshwork observed around beads displaying CDC42 Q61L (Supplementary Fig. 7). These actin structures, however, were composed of branched actin, since their formation was impaired by the CK-666 compound, as measured by their total fluorescence and length. These actin structures depended on the WRC, as shown using extracts prepared from NCKAP1-depleted cells, and did not depend on the CDC42 effector N-WASP, since they were not inhibited by the N-WASP inhibitory compound wiskostatin[46] (Supplementary Fig. 7). Extracts prepared from PPP2R1A-depleted cells did not support actin polymerization from RAC1 Q61L displaying beads (Fig. 7b). Yet the extracts prepared from NHSL1-depleted cells and those prepared from cells simultaneously depleted of PPP2R1A and NHSL1 were as

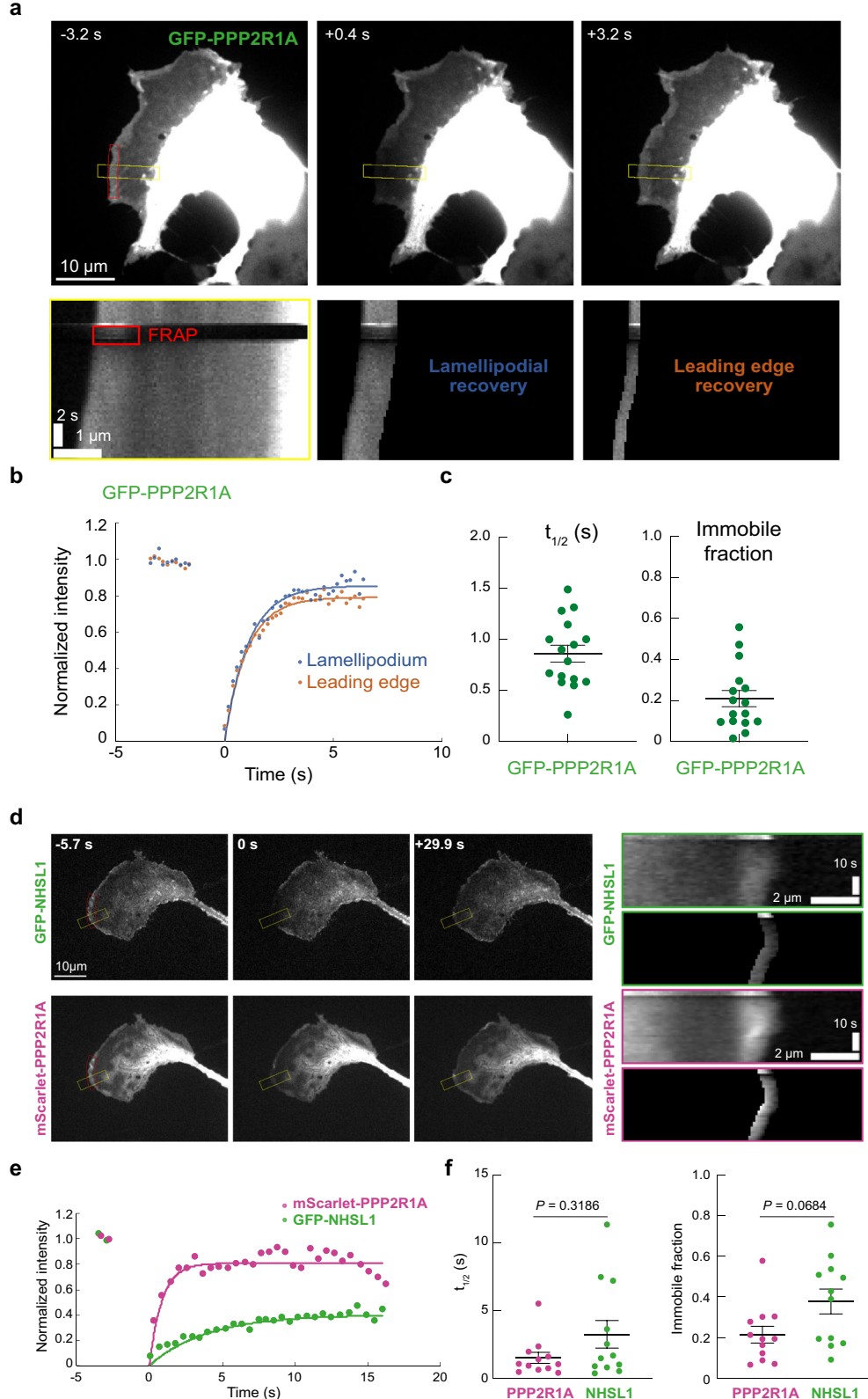

**Fig. 5 | FRAP of PPP2R1A and NHSL1. a** B16-F1 cells transfected with GFP-PPP2R1A were subjected to photobleaching in the red area at time 0. Kymographs built from the yellow rectangle showing the recovery, with filtering for the width of the lamellipodium or the lamellipodial edge. **b** FRAP quantification. **c** Quantification of the time of half recovery and of the immobile fraction in the width of lamellipodia. Mean ± sem, $n = 16$ from three independent experiments. **d** B16-F1 cells were co-transfected with plasmids encoding GFP-NHSL1 and mScarlet-PPP2R1A. The two fluorescent proteins were subjected to simultaneous photobleaching in the red area at time 0. Kymographs built from the yellow rectangle show the recovery of the lamellipodial edge. **e** FRAP quantification. **f** Quantification of the time of half recovery and of the immobile fraction of both fluorophores. Student's *t* test. Mean ± sem, $n = 13$ from three independent experiments. Statistical significance was calculated with two-tailed Mann–Whitney test and *P* values are indicated. Source data are provided as a Source Data file.

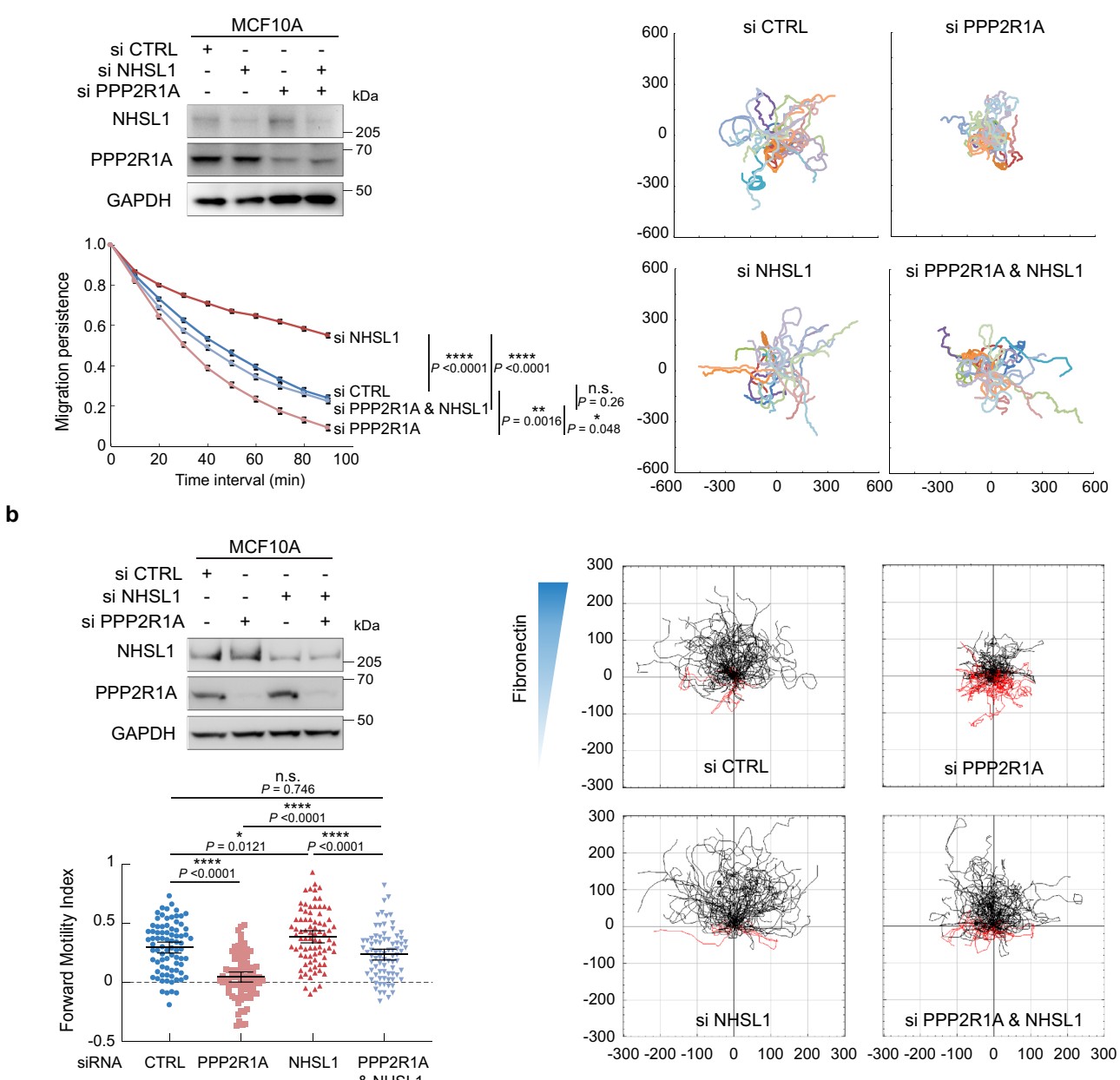

**Fig. 6 | PPP2R1A depends on NHSL1 to regulate cell migration.** MCF10A cells were transfected with pools of siRNAs targeting PPP2R1A, NHSL1 or both. Non-targeting siRNAs were transfected as controls (CTRL). Western blots were used to validate depletion. **a** 2D cell trajectories and migration persistence of single MCF10A cells transfected with indicated siRNAs, Tracking 6 h, $n = 23$ cells. Three biological repeats with similar results, only one is displayed. Data are shown as mean ± SEM. Statistical significance was calculated with custom-made R programs

and $P$ values are indicated. **b** Haptotaxis of MCF10A cells along a gradient of fibronectin. Cell trajectories of MCF10A cells transfected with indicated siRNAs. Three biological repeats with similar results. Tracking 8 h, $n = 81$ cells. Mean ± 95% confidence intervals of Forward Motility Index are plotted. Trajectory plots are labeled in μm. Statistical significance was calculated with ordinary one-way ANOVA with post hoc Tukey's multiple comparison test and $P$ values are indicated. $*P < 0.05$; $**P < 0.01$; $****P < 0.0001$. Source data are provided as a Source Data file.

competent as wild type extracts in supporting actin polymerization. NHSL1 was detected associated with GST-RAC1 Q61L beads (Supplementary Fig. 8), in line with the fact that both CYFIP1 and NHSL1 harbor binding sites for active RAC1[9,19].

We were struck by the fact that RAC1 Q61L suppressed the effect of PPP2R1A depletion in cell migration but not in the in vitro assay. However, we figured that the NSC23766 inhibitor that prevents RAC1 activation by guanine exchange factors blocked actin polymerization in a dose-dependent manner (Fig. 7c), suggesting that endogenous RAC1 activity was required for actin polymerization downstream of

RAC1 Q61L. This result was confirmed using siRNA-mediated depletion of RAC1 from cell extracts (Fig. 7d). Together these observations corroborate the positive feedback loop hypothesis, since constitutively active RAC1 activates endogenous RAC1, and show the importance of PPP2R1A when this positive feedback is required. Interestingly, three independent phosphatase inhibitors that blocked PP2A activity were inactive in the in vitro assay, reinforcing the suggestion that the role of PPP2R1A in regulating migration persistence and actin polymerization does not rely on PP2A phosphatase activity.

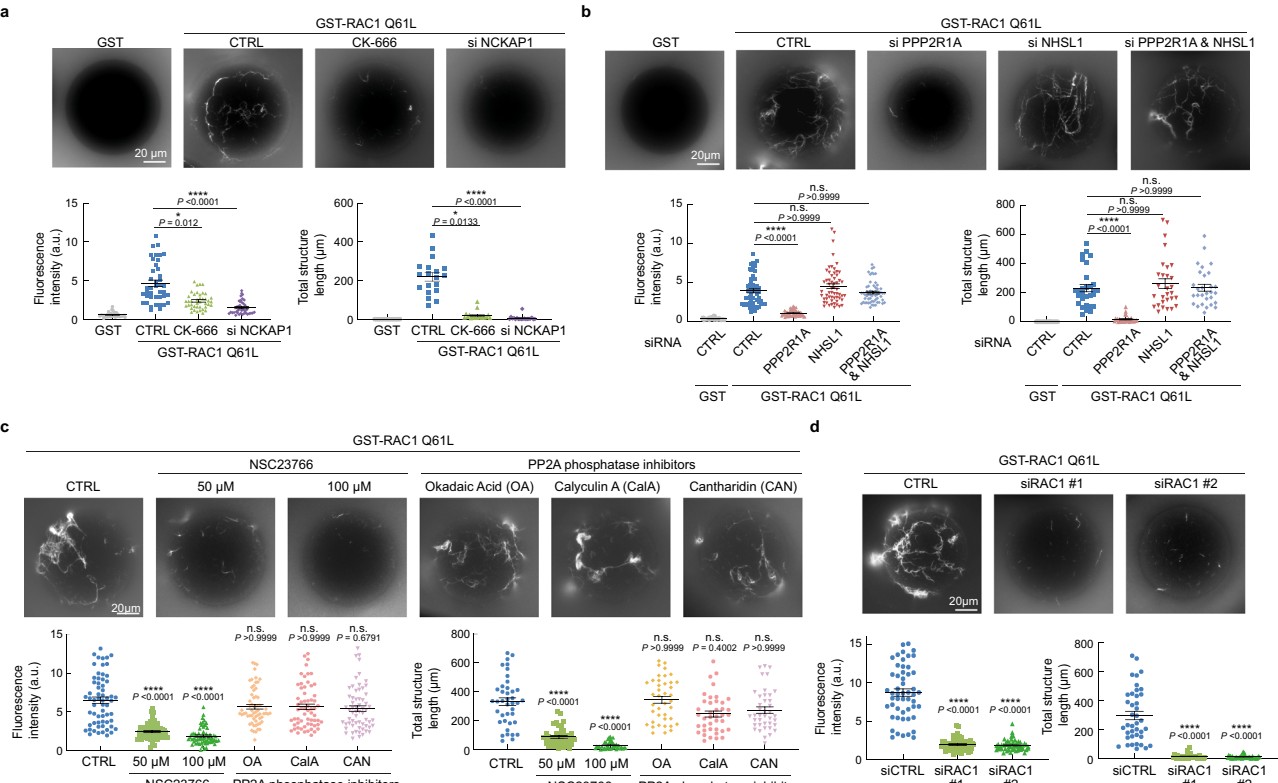

**Fig. 7 | WSC-dependent polymerization of branched actin in cell-free extracts.**
Beads coated with GST or GST-RAC1 Q61L were incubated with extracts of MCF10A cells. Rhodamine-labeled actin structures polymerized on the beads were examined by fluorescent microscopy and quantified by average fluorescence intensity and total structure length. All drug treatments were for 1 h and were controlled with DMSO. Three biological repeats with similar results for all experiments. *n* corresponds to the total number of beads used for quantification from all three experiments. Data are shown as mean ± SEM. **a** Cells were treated with 200 μM CK666 or depleted from NCKAP1 before preparation of extracts. *n* = 41 for fluorescence intensity, *n* = 19 for structure length. **b** Extracts were prepared from MCF10A cells depleted of PPP2R1A, NHSL1 or both with siRNAs. *n* = 58 for

fluorescence intensity, *n* = 29 for structure length. **c** Extracts were prepared from MCF10A cells treated with the NSC23766 inhibitor of RAC1 activation or with the phosphatase inhibitors reported to inactivate PP2A phosphatase activity, okadaic acid (OA, 30 nM), calyculin A (CalA, 10 nM), or cantharidin (CAN, 10 μM). *n* = 62 for fluorescence intensity, *n* = 40 for structure length. **d** Extracts were prepared from MCF10A cells depleted of RAC1 with siRNAs. *n* = 52 for fluorescence intensity, *n* = 40 for structure length. Statistical significance was calculated with Kruskal-Wallis test with post hoc Dunn's multiple comparison test (**a**–**c**) or two-tailed unpaired t-test (**d**) and *P* values are indicated. **P* < 0.05; *****P* < 0.0001. Source data are provided as a Source Data file.

## Uncoupling PPP2R1A from WSC impairs migration persistence

To uncouple PPP2R1A from WSC, we mapped PPP2R1A binding sites in NHSL1. NHSL1 was divided in four fragments excluding the WHD. Upon transient transfection of 293T cells with GFP fusion of NHSL1 fragments, we found that the 4th fragment at the C-terminus of NHSL1 retrieved a large amount of PPP2R1A upon GFP immunoprecipitation (Fig. 8a). The 2nd fragment of NHSL1 that contains the two previously reported binding sites for the SH3 domain of ABI1[19] also retrieved PPP2R1A, but to a much lesser extent than the 4th fragment. The AlphaFold2 software predicted with high confidence three motifs in fragment 4 that interact with the PPP2R1A scaffold or regulatory subunits retrieved in the WSC TAP (Fig. 8b). The first two motifs recognize the first HEAT repeat of PPP2R1A, but with different binding and probably mutually exclusive modes. The third motif of NHSL1 fragment 4 corresponds to a previously described consensus motif for binding regulatory subunits of the B56 family, such as PPP2R5D and PPP2R5E[47]. Upstream of the consensus motif, binding is reinforced by a long stretch of 25 residues containing two successive proline residues (Supplementary Fig. 8). We isolated MCF10A cells that stably express NHSL1 fragment 4. Migration persistence was dramatically decreased in these cells compared to controls (Fig. 8c, Supplementary Fig. 9, and Supplementary Movie 11), suggesting that coupling PPP2R1A to the WSC is critical to regulate cell migration. To obtain further evidence of this point, we investigated the activity of mutated PPP2R1A.

*PPP2R1A* is frequently mutated in tumors, especially in gynecologic cancers[48]. We then isolated stable transfectants from the *PPP2R1A* KO cell line that expressed FLAG-GFP PPP2R1A fusion proteins, either wild type or harboring the frequent cancer-associated mutations, P179R, R183W, or W257C. GFP immunoprecipitations showed that the three mutations strongly impair binding to the WSC, but not to the PP2A catalytic subunit (Fig. 9a). In the 2D cell migration assay, *PPP2R1A* KO MCF10A cells displayed reduced migration persistence, like PPP2R1A knocked-down cells. Reduced persistence was associated with decreased RAC1 levels and activity (Supplementary Fig. 1). Whereas wild-type PPP2R1A rescued the phenotype, mutated forms of PPP2R1A were unable to restore persistence (Fig. 9b and Supplementary Movie 12). This experiment confirms that the interaction of PPP2R1A with the WSC is important for migration persistence.

We then differentiated acini from the stable rescued cell lines at the surface of Matrigel. *PPP2R1A* KO MCF10A cells were impaired in their ability to differentiate hollow acini with correct cell polarity, a hallmark of cell transformation (Fig. 9c). The multicellular structures formed instead were also less regularly spherical than controls. These phenotypes were rescued by wild type PPP2R1A, but not with the derivatives harboring a mutation. These results confirm that the tumor-associated mutations are loss-of-function mutations and suggest that the interaction of PPP2R1A with the WSC might be important to prevent cancer progression.

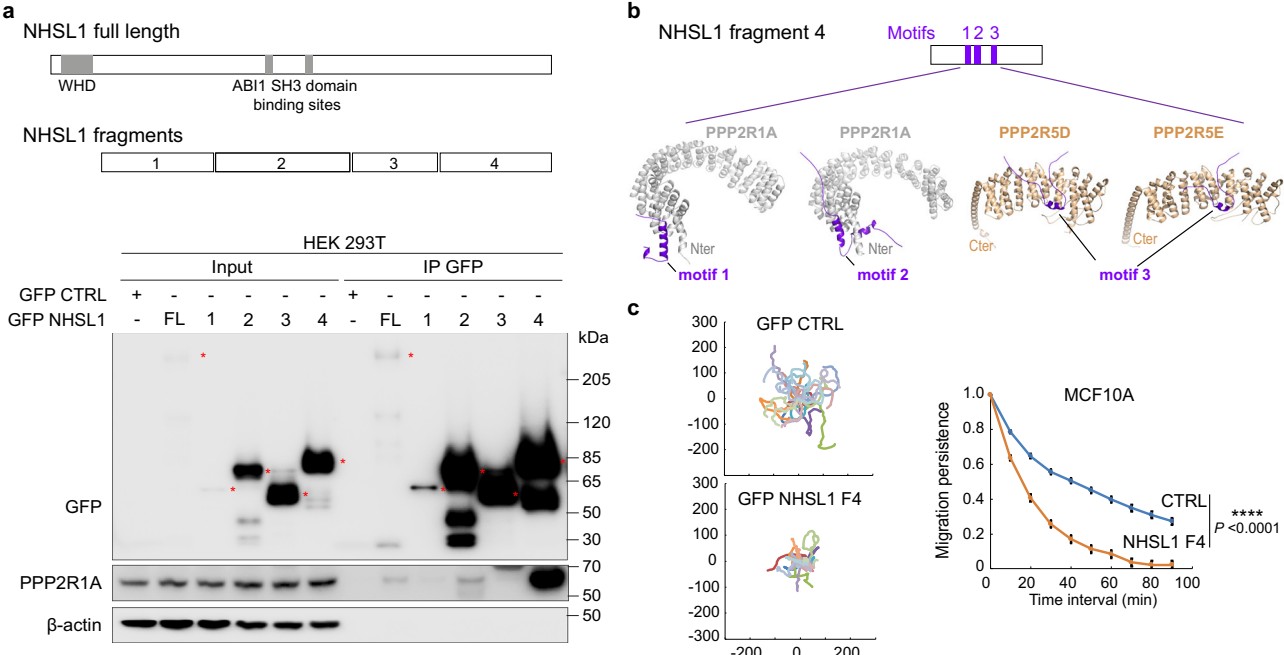

**Fig. 8 | The C-terminal fragment 4 of NHSL1 that binds to PPP2R1A is dominant-negative towards migration persistence. a** Scheme of NHSL1 FL and fragments. WHD: WAVE homology domain. HEK 293 T cells were transiently transfected with GFP-NHSL1 FL or NHSL1 fragments. Transfected cells were subjected to GFP immunoprecipitation. Western blots with the indicated antibodies. Three biological repeats of the same experiment with similar results, only one is displayed. **b** Prediction of three structural motifs that recognize the scaffolding subunit PPP2R1A or the regulatory subunit PPP2R5D of the PP2A complex using Alpha-Fold2. Cartoon representations of the complexes between PPP2R1A (gray) and

motif 1($P_{1380}$SRP-DDH$_{1410}$ of NHSL1 in violet or motif 2 ($A_{1430}$SP-EPS$_{1490}$ of NHSL1 in violet). Motif 3 ($S_{1522}$LS-EPV$_{1569}$ of NHSL1 in violet) interacts with PPP2R5D (wheat) and PPP2R5E (wheat). **c** Cell trajectories, migration persistence, cell speed and MSD were extracted from 2D random migration of MCF10A cells stably expressing GFP-NHSL1 fragment 4. Tracking 7.5 h, $n = 23$ cells. Trajectory plots are labeled in μm. Two biological repeats of the same experiment with similar results, only one is displayed. Data are shown as mean ± SEM. Statistical significance was calculated with custom-made R programs and $P$ values are indicated. ****$P < 0.0001$. Source data are provided as a Source Data file.

## Discussion

In this study, we uncovered a regulator of migration persistence, PPP2R1A, using differential proteomics when RAC1 was activated and when the formation of its downstream product, branched actin, was impaired. PPP2R1A is a subunit of the PP2A phosphatase complex, but we found no evidence that PP2A phosphatase activity was involved in PPP2R1A-dependent regulation of migration persistence: (i) in differential proteomics, levels of catalytic subunits retrieved in ABI1 TAP did not vary as those of PPP2R1A; (ii) we were unable to identify a site on the RAC1/WRC/WSC machinery, whose phosphorylation would depend on PPP2R1A; (iii) PPP2R1A-dependent actin polymerization downstream of RAC1, reconstituted in a bead assay, did not require PP2A phosphatase activity in the cell extract. It is thus possible that the function of PPP2R1A in regulating migration persistence is independent from the PP2A complex. In line with this idea, a large-scale proteomics-based quantification of protein abundance has previously revealed that PPP2R1A was almost one order of magnitude more abundant than PP2A catalytic and regulatory subunits with which it associates[49]. Yet there is no mutual exclusivity of PPP2R1A interaction with the WSC and assembly of PP2A complexes, since we detected PP2A catalytic and regulatory subunits when we selected the WSC by TAP.

The A-type scaffold subunit PPP2R1A is composed of 15 HEAT repeats, onto which the catalytic C subunit and one of the various B regulatory subunits assemble to form the ABC holoenzyme of PP2A[32]. PPP2R1A is mutated in various cancers, particularly often in endometrial cancers (up to 20% in some subtypes)[36]. Tumor-associated PPP2R1A substitutions fall into hotspots. Most point mutations cluster in HEAT repeats 5 and 7, where they have been shown to impair the interaction with B regulatory subunits to a varying extent depending on the mutation and the B subunit considered[50,51]. The 3 mutations of

PPP2R1A, P179R, and R183W in HEAT repeat 5 and W257C in HEAT repeat 7, strongly impaired the interaction with the WSC. This result highlights the importance of the indirect binding mode of PPP2R1A to NHSL1 through NHSL1 motif 3 and B regulatory subunits[47]. The first two motifs of NHSL1 that directly bind to PPP2R1A might contribute to an avidity effect, involving simultaneous direct and indirect interactions, required for a strong interaction of the WSC with PPP2R1A. One of the 3 mutations we study here in the breast cell line MCF10A, R183W, has been detected in 7% of mammary carcinomas[52]. Using a *PPP2R1A* knock-out cell line we examined the rescue provided by wild type or mutated forms. We unambiguously showed that these tumor-associated mutations inactivate PPP2R1A with respect to migration persistence and to the differentiation of polarized acini in Matrigel. Impairing cell polarity in acini is clearly relevant for cancer progression[53,54]. On the contrary, one can expect at least some persistence of tumor cell migration to be required for tissue invasion and metastasis formation. *PPP2R1A* is an atypical tumor suppressor gene, since only one of the two alleles is mutated in tumors, which is classical of oncogenes. Haploinsufficiency and dominant negative effects are thought to account for PPP2R1A inactivation in tumors[50,55].

The here uncovered WSC, where NHSL1 replaces the WAVE subunit, is an alternative form of WAVE complex that was suspected, based on the presence of a WAVE homology domain (WHD) in Nance-Horan Syndrome family proteins[22], but was not previously reported. Now that we have identified the NHSL1-containing WSC, it would be interesting to know whether other proteins of the family, such as NHS and NHSL2, can form similar WAVE shell complexes. The question is difficult to address, since it took the identification of another protein, PPP2R1A, to isolate the NHSL1-containing WSC from other NHSL1 conformations bound to the WRC. It was previously reported[19], and has been confirmed here, that NHSL1 can bind to the WAVE containing WRC

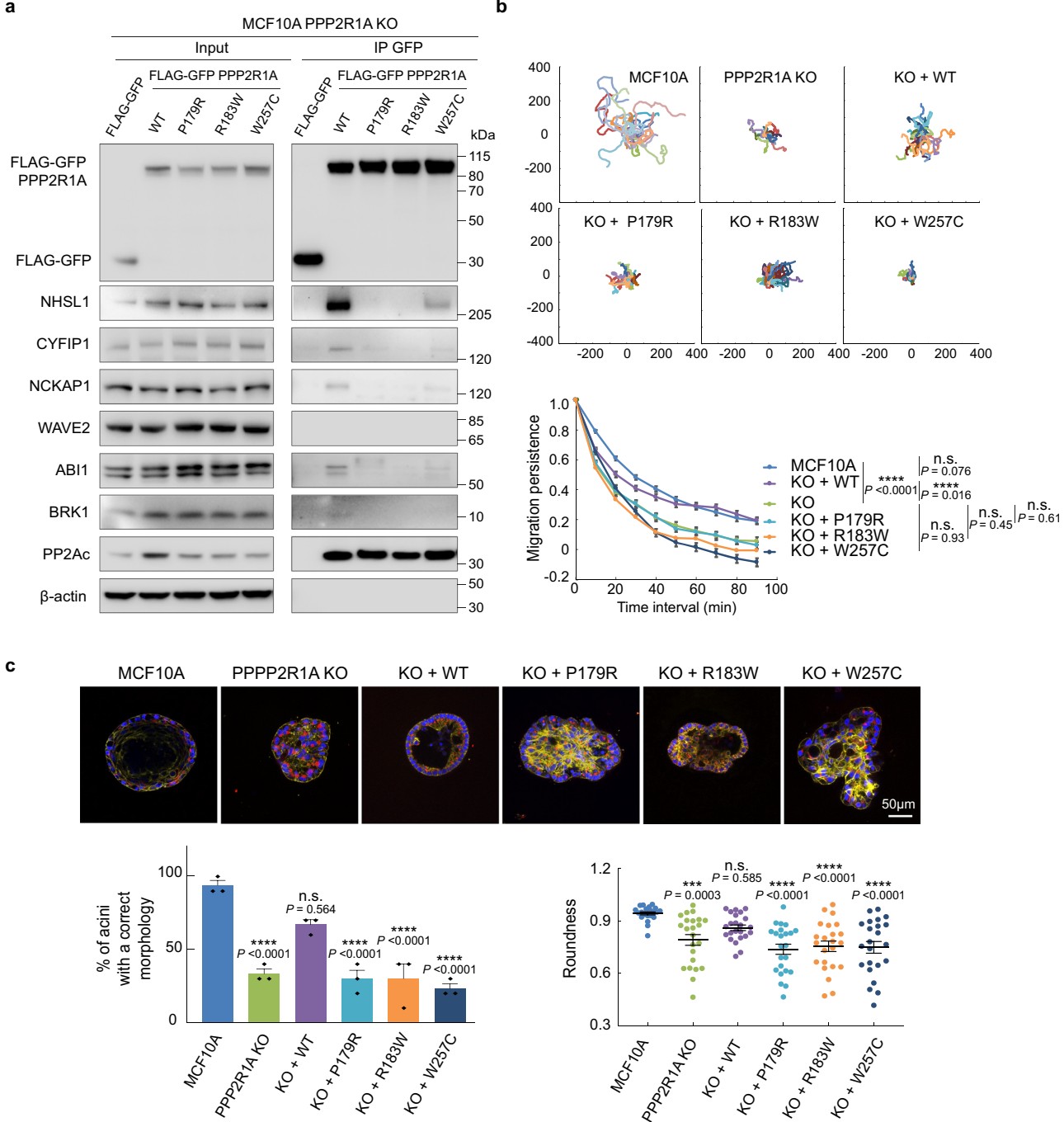

**Fig. 9 | Effects of tumor-associated mutations of PPP2R1A. a** Impairment of WSC interaction. *PPP2R1A* knock-out MCF10A cells stably transfected with plasmids expressing GFP-tagged WT or mutant forms of PPP2R1A from tumors of patients were subjected to GFP immunoprecipitation and immunoprecipitates were analyzed by western blots with antibodies that recognize WRC, WSC, and the catalytic subunit of the PP2A phosphatase complex. Three biological repeats with similar results. **b** Impairment of migration persistence. 2D cell trajectories and migration persistence of MCF10A parental cells, KO, and rescue derivatives. Three biological repeats with similar results, only one is displayed. Tracking 8 h, *n* = 19 cells. Data are shown as mean ± SEM. Statistical significance was calculated with custom-made R programs and *P* values are indicated. **c** Impairment of epithelial morphogenesis. MCF10A parental cells, *PPP2R1A* KO and rescue derivatives were seeded onto Matrigel and grown for 3 weeks. Acini were fixed and stained with DAPI (blue), phalloidin (green) and antibodies targeting the apical Golgi marker GM130 (red). Multicellular structures were observed using confocal microscopy. Three biological repeats with similar results. Acini with altered polarity or no lumen were scored as abnormal. Roundness of acini outlines, based on central confocal sections. Quantification of the three biological repeats. *n* = 23 acini. Data are shown as mean ± SEM. Statistical significance was calculated with Kruskal–Wallis test with post hoc Dunn's multiple comparison test (left) or ordinary one-way ANOVA with post hoc Tukey's multiple comparison test (right) and *P* values are indicated. \**P* < 0.05; \*\**P* < 0.01; \*\*\**P* < 0.001; \*\*\*\**P* < 0.0001. Source data are provided as a Source Data file.

through its C-terminus, which is recognized by the ABI1 SH3 domain. A particularly striking observation is that PPP2R1A is required for migration persistence, whereas NHSL1 inhibits persistence. Importantly, each protein depends on the other to perform its function. PPP2R1A might suppress the negative role of NHSL1 in the WSC. It can also be that the NHSL1-containing WSC has a positive regulatory role, whereas the other NHSL1 pool inhibits the WRC. The identification of the WSC raises several questions. Does the WSC correspond to an intermediate in the WRC life cycle? If so, at which stage? During WRC assembly or during the activation of an already assembled WRC? Is it possible that the NHSL1-containing WSC is just a parallel assembly alternative to the WRC?

Our main findings that PPP2R1A regulates migration persistence through its ability to interact with the NHSL1-containing WSC appears counter-intuitive, because the WSC does not contain WAVE and thus cannot directly activate Arp2/3. Nonetheless, the data suggests that the PPP2R1A-WSC interaction can contribute to WRC activation. That would be in addition to the well-established primary mechanism, which has been reconstituted in vitro without PPP2R1A and NHSL1: active RAC1 activates the WRC through a conformational change that exposes the Arp2/3 activating WCA domain of WAVE[9,12]. Freeing WAVE from the WRC that maintains it inactive[6,7] would be another way to expose its Arp2/3-activating WCA motif. WAVE proteins were reported to be degraded by proteasomes upon cell activation and ubiquitylated on a specific lysine residue in their WHD[56,57]. But these reports do not demonstrate that this is a free form of WAVE that is degraded after activation. Exchange events between WAVE and NHSL1 subunits in the shell complex are technically challenging to detect in the cell. Future work should be devoted to this question if we want to refine our understanding of the molecular mechanisms by which PPP2R1A coupling to the NHSL1-containing WSC contributes to migration persistence.

Migration persistence depends on positive feedback[23], which is manifested in propagating waves of branched actin polymerization[25]. Positive feedback has been observed at multiple molecular levels. The simple fact that the product of the Arp2/3 reaction is an actin filament that can in turn become the substrate of another Arp2/3-dependent branching reaction is positive feedback referred to as an autocatalytic reaction[58]. The WRC turns over at the lamellipodium edge due to elongation of actin filaments that it contributes to generating[27,59]. Coronin1A decorates lamellipodial actin and further activates RAC1 via its association with ArhGEF7[60]. Similarly, we found PPP2R1A in the width of the lamellipodium, ideally localized to sense lamellipodial actin, whereas PPP2R1A association with the WSC is likely to take place at the lamellipodium edge, where NHSL1 is localized and turns over with similar kinetics as PPP2R1A. The interaction of PPP2R1A with WSC would thus be where Arp2/3 generates branched actin (Fig.10).

Many observations converge on a potential implication of PPP2R1A in a positive feedback loop: (i) we initially focused on PPP2R1A, among the many ABI1 partners, because its interaction with the WSC was regulated by branched actin; (ii) PPP2R1A depletion by knock-down and knock-out decreases RAC1 activity and levels; (iii) PPP2R1A is dispensable for migration persistence when RAC1 is constitutively activated by the Q61L mutation; and (iv) the requirement for PPP2R1A in the polymerization of branched actin is associated with positive feedback where RAC1 Q61L immobilized at the surface of beads activates endogenous RAC1 from the extract. All of this evidence points to a possible role for PPP2R1A, and the WSC it associates with, in the positive feedback that sustains directional migration through continuous actin polymerization at the leading edge. Future work should aim to find ways to dissect, at the molecular level, the complex circuitry that mediates feedback and persistence.

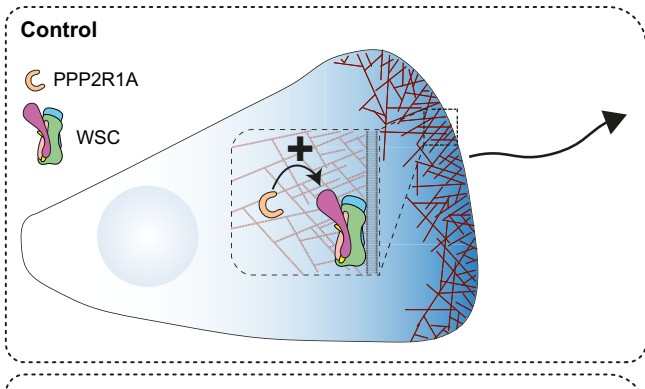

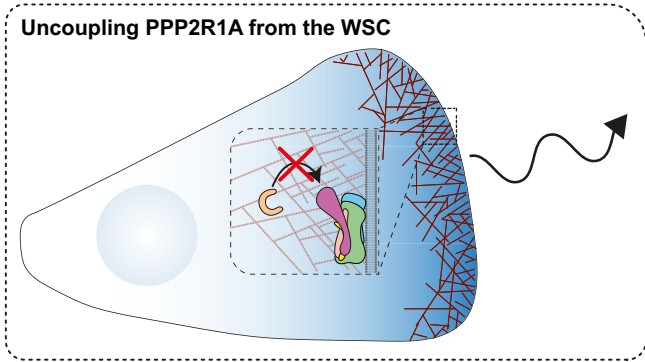

**Fig. 10 | Model.** PPP2R1A, localized throughout the lamellipodium width, interacts with the WSC at the lamellipodial edge to mediate migration persistence. When PPP2R1A is uncoupled from the WSC using the fragment 4 of NHSL1 or tumor-associated mutations, migration persistence decreases.

## Methods

### Cells and drugs

MCF10A cells were grown in DMEM/F12 medium supplemented with 5% horse serum, 20 ng/mL epidermal growth factor, 10 µg/mL insulin, 100 ng/mL cholera toxin, 500 ng/mL hydrocortisone, and 100 U/mL penicillin/streptomycin. MDA-MB-231, HEK 293 T and B16-F1 cells were grown in DMEM medium with 10% FBS and 100 U/mL penicillin/streptomycin. Medium and supplements were from Life Technologies and Sigma. Cells were incubated at 37 °C in 5% CO2.

B16-F1 (CRL-6323) and HEK 293 T (CRL-3216) cells were from ATCC. MCF10A and MDA-MB-231 cell lines were from the collection of breast cell lines organized by Thierry Dubois (Institut Curie, Paris). MCF10A RAC1 Q61L cell line was bought from Horizon Discovery Ltd. (Cambridge, UK).

CK-666, NSC23766 and wiskostatin were purchased from Merck. The three PP2A inhibitors, okadaic acid sodium salt, CalyculinA and Cantharidin, were purchased from Santa Cruz Biotechnology.

### Plasmids and in vitro translation

The following plasmids were built using a building block method:[61]

MXS AAVS1L SA2A Puro bGHpA EF1FLAG GFP Blue2 Sv40pA AAVS1R (control)

MXS AAVS1L SA2A Puro bGHpA EF1 FLAG GFP ABI1 Sv40pA AAVS1R

MXS AAVS1L SA2A Puro bGHpA EF1 FLAG GFP PPP2R1A Sv40pA AAVS1R

MXS AAVS1L SA2A Puro bGHpA EF1 FLAG GFP ARPC1B Sv40pA AAVS1R

MXS PGK Blasti bGHpA CAG PC HA Blue2 Sv40pA (control)

MXS PGK Blasti bGHpA CAG PC HA BRK1 Sv40pA

MXS PGK ZeoM bGHpA EF1Flag mScarlet PPP2R1A SV40pA

The plasmid pCAG-EGFP-NHSL1-IRES-Puro encodes a GFP fusion protein with FL NHSL1 (1-1639) including the WHD domain. FL NHSL1

corresponds to the isoform X6 (XP_047275069.1) with 7 substitutions T440I, A449V, T891A, G1082E, A1123T, P1329S, and S1466N. The fragments 1 (MVVFI…KSRDH), 2 (LISRH…EGSGT), 3 (MKKLD…VEPAE), and 4 (NVSEA…LSEES) correspond to the isoform X19 (XP_047275073.1) with 7 substitutions T363I, A372V, T814A, G1005E, A1046T, P1252S, and S1389N and were expressed as GFP fusion proteins from the same plasmid as the full-length NHSL1. The MBP - NHSL1 (28–172) fusion protein with a TEV cleavage site between the two proteins was expressed from a modified pMAL-c2X (New England Biolabs).

For in vitro translation, ORFs were subcloned into pCS2 or pCS2-HA, and expressed from the SP6 promoter in the TnT® Quick Coupled Transcription/Translation System (Promega L2080) in the presence of $^{35}$S-Met (PerkinElmer NEG709A005MC), as described previously[5,43].

## Stable cell lines
Stable transfections of MCF10A and MDA-MB-231 cells were performed using Lipofectamine 3000 (Invitrogen) with the plasmids encoding FLAG-GFP, FLAG-GFP ABI1, FLAG-GFP PPP2R1A, GFP NHSL1 Fragment 4. To obtain stable integration at the AAVS1 site, cells were co-transfected with two TALEN constructs (Addgene #59025 and 59026) inducing a double strand break at the AAVS1 locus[62]. Cells were selected with 1 μg/mL puromycin (Invivogen) and pooled.

Stable MCF10A double cell line was obtained by transfecting cells stably expressing FLAG-GFP PPP2R1A with PC-HA BRK1 or PC-HA Blue2. Cells were then selected with 10 μg/ml of Blasticidin (Invivogen). Single clones were expanded and analyzed by Western blot.

## Knockdown and knockout
MCF10A and MDA-MB-231 knockdown cells were obtained by transfecting 20 nM siRNAs with Lipofectamine RNAiMAX (Invitrogen). siRNAs were Dharmacon ON-TARGET SMART Pools (L-010259-00-0010 for PPP2R1A, L-032698-00-0010 for NHSL1, L-010640-00-0010 for NCKAP1) or Sigma siRNAs CCTTTGTACGCTTTGCTCA and GCCAC-TACAACAGAATTTT to target RAC1. Cells were analyzed after 3 days.

MCF10A knockout cell lines were generated with CRISPR/Cas9 system. The following gRNAs were used:
*PPP2R1A* 5′-CATAGACGAACTCCGCAATG-3′
*NHSL1* 5′-TCGGCTTTCCTCATCTAGGT-3′
non-targeting 5′-AAAUGUGAGAUCAGAGUAAU-3′.

Cells were transfected with gRNA, tracrRNA, and Cas9 protein by Lipofectamine CRISPRMAX™ (all reagents from ThermoFischer Scientific). After 2 days, cells were diluted at 0.8 cells/well in 96-well plates. Single clones were expanded and analyzed by western blot. The positive clones were confirmed by sequencing.

## Antibodies
The antibodies used were: anti-PPP2R1A (Bethyl Laboratories, A300-962A, 1:2000); anti-NHSL1 (Sigma-Aldrich, HPA029967, 1:1000); anti-PP2Ac (Bethyl Laboratories, A300-732A, 1:3000); anti-GFP (Roche, 11814460001, 1:1000); anti-NCKAP1 (Bethyl Laboratories, A305-178A, 1:3000); anti-WAVE1 (R&D Systems, AF5514, 1:1000); anti-WAVE3 (R&D Systems, AF5515, 1:1000); anti-phospho-WAVE2 Ser308 (Millipore, 07-1511, 1:1000); anti-RAC1 (BD Biosciences, #610651, 1:500); anti-phospho-RAC1 Ser 71 (Cell Signaling Technology, #2461 T, 1:1000); anti-GAPDH (Thermo Fisher Scientific, AM4300, 1:4000); anti-β-actin (Thermo Fisher Scientific, AM4302, 1:4000); and anti-GM130 (#610823, BD Biosciences, 1:100). Home-made CYFIP1, ABI1, WAVE2 antibodies and BRK1 antibody were described previously[5,43].

## Western blots
Cells were lysed in XB-NP40 buffer (50 mM HEPES, 50 mM KCl, 1%NP-40, 10 mM EDTA, pH 7.7) supplemented with protease inhibitor (Roche), the lysates were clarified by centrifugation at 13,000 rpm for 15 min and subjected to SDS−PAGE using NuPAGE 4-12% Bis-Tris or 3–8% Tris-Acetate gels (Life Technologies). Pieces of nitrocellulose membranes were cut and incubated with different primary antibodies, HRP conjugated secondary antibodies (Sigma) and developed with SuperSignal™ West Femto Substrate (Thermo Fisher Scientific) and ChemiDoc imaging system (BIO-RAD). Densitometry of Western blots was performed with ImageJ. Uncropped images are provided in the source data file.

## Immunoprecipitations and tandem affinity purification of the WSC
Cells stably expressing FLAG-GFP ABI1 or FLAG-GFP PPP2R1A were lysed with XB-NP40 buffer (50 mM HEPES, 50 mM KCl, 1% NP-40, 10 mM EDTA, pH 7.7) supplemented with protease inhibitors at 4 °C for 30 min. The phosphatase inhibitor cocktail PhosSTOP (Roche) was added for phosphosite analysis. The lysates were clarified by centrifugation at 13,000 rpm for 15 min. Clarified cell extracts were incubated with FLAG-M2 beads (Sigma) at 4 °C for 4 h. FLAG-M2 beads were washed with XB-NP40 buffer and eluted with 0.5 mg/ml FLAG peptide (Sigma) in XB (50 mM HEPES, 50 mM KCl, 10 mM EDTA, pH 7.7) overnight at 4 °C. FLAG elutions were collected and incubated with GFP-trap beads (Chromotek) at 4 °C for 1 h. The GFP-trap beads were washed with XB-NP40 buffer. 20% of the beads were subjected to SDS−PAGE for Western blot or silver staining (SilverQuest Silver Staining Kit, Thermo Fisher Scientific). In all, 80% of the beads were analyzed by mass spectrometry.

In vitro translation reactions of 50 μl were diluted with 450 μl of XB buffer (20 mM Hepes, 100 mM KCl, 1 mM MgCl2, 0.1 mM EDTA, pH 7.7) and 10 μl of beads coupled to an anti-HA mAb (Merck, A2095) were added. The mixture was incubated for 2 h at 4 °C. Then the beads were washed four times with 1 ml of XB buffer and resuspended in SDS loading buffer. SDS−PAGE gels were incubated with gel drying buffer (15% methanol and 5% glycerol) for 1 h, then dried. Autoradiography was revealed with a Typhoon imager FLA 7000 (GE Healthcare).

To purify the WSC, MCF10A cells stably expressing FLAG-GFP PPP2R1A and PC-HA BRK1 were lysed with XB-NP40 buffer (50 mM HEPES, 50 mM KCl, 1% NP-40, 10 mM EDTA, pH 7.7) supplemented with protease inhibitors at 4 °C for 1 h, then the lysates were clarified by centrifugation at 13,000 rpm for 15 min. Cell extracts were incubated with FLAG-M2 beads (Sigma) overnight at 4 °C. FLAG-M2 beads were washed with XB-NP40 buffer, and eluted with 0.5 mg/ml FLAG peptide (Sigma) in FLAG-elution buffer (50 mM HEPES, 50 mM KCl, 1 mM CaCl$_2$, pH 7.7). FLAG elutions were incubated with PC beads (Anti-Protein C Affinity Matrix, Sigma) overnight at 4 °C. PC beads were washed and eluted with EGTA-containing elution buffer (50 mM HEPES, 50 mM KCl, 10 mM EGTA, pH 7.7) overnight. PC elutions were subjected to SDS−PAGE and mass spectrometry.

## Purification of recombinant proteins and complexes
GST and GST fusion proteins with RAC1 WT, RAC1 Q61L, CDC42 Q61L and the RAC1 binding domain of PAK1 were expressed in and purified from *E. coli* BL21. After 3 hours induction at 37 °C with 1 mM IPTG, cells were resuspended and lysed in lysis buffer (50 mM Tris-HCl, 100 mM NaCl, 2.5 mM CaCl$_2$, 10 mM MgCl$_2$, 1% Triton X100, 5% glycerol, 0.5 mg/ml lysozyme, 10 μg/ml DNaseI, 1 mM DTT, pH 8.0, protease inhibitors) at 4 °C for 1 h. The lysates were clarified by centrifugation at full speed in a tabletop centrifuge for 30 min. The supernatants were incubated with Glutathione Sepharose beads (GE healthcare) at 4 °C for 3 h. The beads were washed with TBS buffer (50 mM Tris, 100 mM NaCl, 2.5 mM CaCl$_2$, 5 mM MgCl$_2$, 1 mM DTT, pH 8.0), then the bound proteins were eluted with GST elution buffer (50 mM Tris-HCl, 5 mM MgCl$_2$, 10 mM glutathione, pH 8.0). The elutions were dialyzed in buffer (50 mM Tris, 5 mM MgCl$_2$, 20% glycerol, pH 8.0) and kept at −80 °C until use.

To purify the recombinant WSC, the protocol for recombinant WRC was followed[44]. Briefly, CYFIP1 and NCKAP1 were individually expressed as His tagged fusion proteins in insect cells and a dimer

reconstituted; BRK1, ABI1 (1-158), and WAVE1 (1-230) or NHSL1 (28-172) were individually expressed as MBP-fusion proteins in *E.coli* and a trimer reconstituted. Reconstituted pentamers were purified on a SOURCE 15Q (Cytiva) anion exchange column and polished on a Superdex 200 (Cytiva) gel filtration column.

For electron microscopy examination, purified WSC and WRC were diluted to 24 and 33 ng/µl, respectively, in 10 mM Hepes pH 7.0, 100 mM NaCl, 2 mM MgCl$_2$, 2 mM DTT and 10 % glycerol and 3 µl was then applied to freshly glow discharged carbon-coated 200 mesh copper support EM grids (EMS). After 1 min incubation, the sample was blotted and immediately replaced with the stain solution containing 2% uranyl acetate, which was blotted and replaced twice. The grids were imaged on a JEOL 2100 STEM using a Gatan One View 4 K digital camera at 200 keV, 80 ×900 magnification to collect 132 WSC and 194 WRC micrographs at 2 s (-25 e⁻/Å²) exposure with a pixel size of 1.85 Å. The micrographs were processed with cryoSPARC[63] to obtain 2D class averages.

### GST pull-down and RAC1 activation assay

MCF10A cell extracts were prepared with XB-NP40 buffer as described above. 20 µg purified GST fusion proteins were incubated with 20 µl Glutathione Sepharose beads (GE healthcare) in 500 µl incubation buffer (50 mM Tris-HCl, 100 mM NaCl, 2.5 mM CaCl$_2$, 10 mM MgCl$_2$, 1% Triton X100, 5% glycerol, 1 mM DTT, pH 8.0) at 4 °C for 1 h. The pre-coated beads were washed and incubated with 1 ml MCF10A cell extract at 4 °C for 1 h. The beads were washed with XB-NP40 buffer and subjected to Western blot. RAC1 activation was also measured by ELISA (G-LISA kit BK128 from Cytosketon, Inc.) according to the manufacturer's instructions.

### Actin polymerization in cell-free extracts

MCF10A cells were lysed either by nitrogen cavitation (Parr instruments, 500 Psi for 20 minutes) in buffer (50 mM HEPES, 50 mM NaCl, 5 mM MgCl$_2$, 0.1 mM EDTA, 1 mM DTT, pH 7.7, protease inhibitor), or with NP40 containing buffer (50 mM HEPES, 50 mM KCl, 5 mM MgCl$_2$, 1% NP-40, pH 7.7, protease inhibitor) at 4 °C for 30 min. The NP40 containing extract supported RAC1 Q61L-induced actin polymerization, whereas extract obtained by nitrogen cavitation supported CDC42 Q61L-induced actin polymerization. The extracts were clarified by centrifugation at full speed in a tabletop centrifuge for 15 min. 10 µl clarified cell extract was supplemented with 2 µl energy mix (20 mM ATP, 150 mM creatine phosphate, 20 mM MgCl$_2$, 2 mM EGTA) and 0.75 µl rhodamine-actin (1 mg/ml, Cytoskeleton, Inc.). The mixture was centrifuged 5 min at 13,000 rpm. To trigger the reaction, 1 µl Glutathione Sepharose beads bound to 2 µg GST-RAC1 Q61L or GST-CDC42 Q61L were added to a 10 µl reaction mix. After 1 h incubation at room temperature, the reaction was squashed in between a coverslip and the microscope slide. The coverslip was sealed with melted VALAP (Vaseline-Lanoline-Paraffin). The beads were observed under an inverted microscope (Olympus IX83) with 60x oil objective. Fluorescence intensity and structure length on the surface of the beads were measured using ImageJ.

### Migration and videomicroscopy

Random cell migration assays were performed in µ-Slide eight-well ibidi dishes. For 2D migration assays, cells were seeded on the dishes coated with 20 µg/ml Fibronectin (Sigma). For 3D migration assays, cells were sandwiched between two layers of 2 mg/ml collagen (rat tail collagen type I, Corning). After seeding cells for 24 h, videomicroscopy was performed on an inverted Axio Observer microscope (Zeiss) equipped with a Pecon Zeiss incubator XL multi S1 RED LS (Heating Unit XL S, Temp module, CO2 module, Heating Insert PS and CO2 cover), a definite focus module and a Hamamatsu camera C10600 Orca-R2. Images were acquired every 5 or 10 min for 24 h with ×10

objective for 2D migration, and every 10 min for 48 h with ×20 objective for 3D migration. Individual cells were tracked by ImageJ software-Manual Tracking plug-in. DiPer software was used to analyze the cell migration parameters[64].

Fibronectin gradients were prepared with PRIMO photopatterning system (Alvéole). 35 mm ibidi dishes with glass bottom were treated with plasma for 1 min. PDMS stencils (Alvéole) with three 3 × 3 mm wells were stacked on each plasma-treated dish immediately. PDMS stencil wells were coated with PLL-g-PEG for 1 h and rinsed three times with PBS. Then photoinitiator (PLPP) (Alvéole) was added to the PDMS stencil wells for micropatterning. LEONARDO photopatterning software was used to design the micropatterns (width 645 µm, height 1031 µm with 100% to 0% grayscale gradient). The dishes with PDMS stencils were placed onto the microscope holder (Nikon ECLIPSE Ti2) with PRIMO module, and the patterns were projected to the surface at a UV dose of 1500 mg/mm². PDMS stencil wells were rinsed three times with PBS and coated with 50 µg/ml Fibronectin/Fibrinogen-Alexa647 (Invitrogen) for 30 min at 37 °C. The Fibronectin/Fibrinogen-Alexa647 were only adsorbed on the previously illuminated areas. After rinsing the wells three times with PBS, MCF10A cells were seeded in the coated PDMS stencil wells. Cells were washed once with medium after adhering for 2 h. After 24 h, videomicroscopy was performed on an inverted Axio Observer microscope (Zeiss) with the 10x objective. Images were acquired every 10 min for 24 h. Individual cells were tracked by ImageJ software-Manual Tracking plug-in. The tracks obtained were analyzed by the chemotaxis tool (Ibidi) to extract the FMI values and cell trajectory plots. The FMI values were plotted by GraphPad Prism software as mean ±95% confidence intervals.

B16-F1 cells were transiently transfected with GFP and mScarlet plasmids and analyzed by videomicroscopy after 2 days. Videos were acquired using a confocal laser scanning microscope (TCS SP8, Leica) equipped with a high NA oil immersion objective (HC PL APO 63×/1.40, Leica), a white light laser (WLL, Leica) and controlled by the LasX software. Images were taken every 10 s for -5–10 min. Kymographs were drawn using Multi Kymograph tool in ImageJ. To analyze the localization of proteins, radial line scans were performed and analyzed as described[20].

### Fluorescence recovery after photobleaching

B16-F1 cells were transiently transfected with GFP and mScarlet plasmids and analyzed by FRAP after 1 day of expression. Videos were acquired using an inverted Eclipse Ti-E (Nikon) equipped with a CSU-X1-A1 Nipkow Spinning Disk confocal system (Yokogawa), a 100x APO TIRF oil immersion objective (NA: 1.49, Nikon), a 488 nm, 150 mW (Vortran) and 561 nm, 100 mW (Coherent) lasers, a Quad bandpass 440/40 nm, 521/20 nm, 607/34 nm, 700/45 nm dichroic mirror (Semrock), a Prime 95B sCMOS camera (Photometrics) and an iLas 2 module (GATACA Systems). The set-up was controlled by MetaMorph software version 7.7 (Molecular Devices). For single FRAP experiments, images were acquired at a video streaming rate of 200 ms for -10–20 s. For dual FRAP experiments, images were acquired at a frame rate between 1.1 and 1.4 s for about 30-40 s. Since FRAP experiments were performed in extending lamellipodium of migrating cells, the photobleached ROIs moved forward. We used kymographs to obtain the profile of fluorescence recovery in the moving ROI. Kymographs were drawn using the Multi Kymograph tool in ImageJ. We manually draw on kymographs a line that followed the cell edge, and whose thickness covered the whole photobleached lamellipodium or the leading edge. The line was used to reslice kymograph images along the y axis (time), allowing quantification over time of the recovered fluorescence intensity in the photobleached ROI, which moves forward during cell migration. The fluorescence intensity is normalized by the average fluorescence intensity in the ROI before photobleaching.

## 3D acini

Single cells were seeded on a 1-mm-thick solidified layer of Matrigel (growth factor reduced, Thermo Fisher Scientific, #CB-40230C) in the eight-well glass chamber (Merck, #PEZGS0816) and grown for 3 weeks in MCF10A medium with 1% horse serum, 5 ng/mL EGF and 2% Matrigel. Then the acini were fixed and stained with the indicated antibody, Alexa Fluor™ 555 Phalloidin (A34055, Thermo Fisher Scientific) and DAPI. Images were acquired using a confocal laser scanning microscope (TCS SP8, Leica) equipped with a high NA oil immersion objective (HC PL APO 63 × / 1.40, Leica), a white light laser (WLL, Leica) and controlled by the LasX software.

## Statistics

Migration persistence for individual cells is evaluated based on the exponential decay and plateau fit according to Eq. (1).

$$P = (1 - b) * e^{-\frac{t}{a}} + b \qquad (1)$$

Where, $P$ is the migration persistence, $b$ is plateau value, $t$ is the time interval and $a$ is the decay constant. Then the related statistical analysis was conducted through custom-made R programs, as previously described[65].

For other statistical analysis, GraphPad Prism software and Microsoft Excel were used. The Shapiro–Wilk normality test was performed. Two-tailed unpaired t-test was used for parametric data and Mann–Whitney test was used for non parametric data. ANOVA followed by post hoc Tukey's multiple comparison test was used for parametric data and Kruskal–Wallis test followed by post hoc Dunn's multiple comparison test was used for non parametric data.

Four levels of significance were distinguished: *$P < 0.05$, **$P < 0.01$, ***$P < 0.001$, ****$P < 0.0001$.

## Reporting summary

Further information on research design is available in the Nature Portfolio Reporting Summary linked to this article.

## Data availability

Raw files of the LC-MSMS analyses have been deposited in PRIDE with the accession number PXD031584. Files with the reference number 170414 refer to the TAP purification of FLAG-GFP-ABI1, 181220 to the TAP purification of FLAG-GFP-PPP2R1A, 210415 to the TAP purification of the WSC and 200120 to the identification of phosphosites in the TAP purification of FLAG-GFP-ABI1.

The structural models of (i) the Wave Shell Complex (WSC) composed of NHSL1(1-95), CYFIP1, NCKAP1, BRK1 and ABI2(1-160), (ii) the NHSL1(1382-1410)-PPP2R1A complex, (iii) the NHSL1(1430-1490)-PPP2R1A complex, and (iv) NHSL1(1522-1569)-PPP2R5D(80-530) complex, are available in ModelArchive [modelarchive.org] with the accession codes ma-agzek, ma-ne9d4, ma-sx8ix and ma-rop1i, respectively. Source data are provided with this paper.

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

## Acknowledgements

We thank Vassilis Koronakis for guidance in reconstitution assays using cell extracts. We thank Anna Castro, Gregory Giannone and Emmanuel Derivery for critical reading of the manuscript. This work was supported by grants from Agence Nationale de la Recherche (ANR-20-CE13-0016 and ANR-22-CE13-0041 to A.M.G.), Fondation ARC pour la Recherche sur le Cancer (ARC PJA 2021 060003815 to A.M.G.) and the National Institutes of Health (R35 GM128786 to B.C.). Mass spectrometry equipment was subsidized by Conseil Régional d'Ile-de-France (Sesame No 10022268). The PhD of Y.W. was supported by fellowships provided by Fondation pour la Recherche Médicale and by Fondation ARC pour la Recherche sur le Cancer.

## Author contributions

Y.W. performed most of the experiments and participated in writing of the manuscript. G.C. performed the mass spectrometry under the direction of J.V.; S.R. performed live localizations of fluorescent fusion proteins. R.G. generated the structural models. Y.L., D.J.B., and B.C. obtained the recombinant WSC and negatively stained EM images. M.K. generated NHSL1 expression plasmids. C.D., A.B., and A.I.B. helped to set up the haptotaxis assay. A.M.G. and A.P. have jointly supervised the work and wrote the manuscript.

## Competing interests

The authors declare no competing interests.
