## [Peer review file · Nature Communications]

REVIEWER COMMENTS

Reviewer #1 (Remarks to the Author):

In this paper, the authors identified PPP2R1A to be differentially associated with the WAVE complex subunit ABI1 when RAC1 was activated and downstream generation of branched actin was blocked. PPP2R1A was found to associate with the WAVE Shell Complex (WSC), that contains NHSL1 instead of WAVE proteins as in the canonical WAVE Regulatory Complex (WRC). The authors showed that PPP2R1A was required for cell migration persistence and RAC1-dependent actin polymerization via interaction with NHSL1. In addition, cancer-associated PPP2R1A mutations impaired its WSC binding and migration regulation. The findings provide information to better understand the molecular mechanism by which actin polymerization is modulated. However, some necessary information is still missing, especially the biological consequences of the binding of PPP2R1A to WSC and how such interaction was regulated by signaling.

Introduction

1. In the second paragraph, the authors may consider to provide more comprehensive information about the components of the WRC complex.
2. I would suggest to provide more information regarding PP2A and PPP2R1A in the introduction section.

Results

For the label-free proteomics study,

1. The authors may want to provide more details regarding the data analysis and processing procedure. For example, how was data normalization performed for quantification?
2. The authors identified 89 proteins as ABI1 binding partners. However, it's unclear how they were filtered out. Were all proteins observed in the FLAG-GFP samples regarded as background interference? Or was quantitative comparison performed to remove background proteins? More details should be provided in the method section. In addition, many keratin isoforms were listed as ABI1 binding proteins in Table S1. Keratins are normally considered as contaminants in coIP-MS experiments.
3. More information should be provided in Table S1 such as protein accession number, detection coverage, and number of identified peptides. The information of the control samples (FLAG-GFP) should

be included. Additionally, the quantification results should be provided for all identified proteins together with their P values from statistical analysis.

4. For figure 1b, it would be easier to understand the results by showing the fold change of treatment/control instead of the other way around. For example, MCF10A+CK666/MCF10A and RAC1 Q61L+CK666/RAC1 Q61L. In addition, I would suggest to present the results in column graphs or box plots with the significance levels indicated in the figure. All the detailed information about the quantitative analysis could be provided in Table S1.

5. It seems that different modulators responded quite differently to RAC1 activation and Arp2/3 inhibition. I think the authors should discuss about their observations in more details. For example, the authors mentioned that it was consistent with the literature that more lamellipodin were associated with WRC when RAC1 was activated. Based on the results, such interaction was also enhanced when MCF10A cells were treated with CK666. Is this expected and supported by existing knowledge? Similar to PPP2R1A, lamellipodin also responded differently to CK666 treatment in parental and RAC1 Q61L cells. The authors mentioned that RAC1 Q61L did not depend on the feedback of branched actin for its activity. But it's still unclear to me why CK666 treatment affected the association of PPP2R1A to ABI1 in RAC1 Q61L cells.

6. For figure 1c, the model is a little misleading. It shows the dynamic interaction between PPP2R1A and WRC. However, the authors later showed that PPP2R1A bound to WSC.

For the other parts,

1. The authors treated the cells with CK666 to inhibit Arp2/3. What concentration was used? For how long? Did the author test the effects of different doses? The authors should show how the treatment affect the activity of Arp2/3. How does this treatment affect RAC1?

2. The authors tried to study phosphorylation alteration of the WSC or WRC by TAP purification of ABI1 and MS analysis. In my opinion, this is not a very effective strategy. First, the abundance of proteins obtained by TAP purification is normally low. Second, large experimental variation is expected, therefore small quantitative changes cannot be efficiently detected. Third, phosphopeptides normally exist at much lower levels compared to non-phosphopeptides, therefore difficult to be detected without enrichment. I'm surprised that PPP2R1A bound to WSC in the form of PP2A complex without affecting its phosphorylation. Then, what are the biological consequences of such binding? Not detecting any change does not necessarily mean its phosphatase activity is not involved. The author may consider to perform a large scale quantitative phosphoproteomics analysis with proper enrichment and quantification to seek more clues. I am also wondering whether inhibiting the phosphatase activity of PP2A would affect the functions of PPP2R1A in the context of migration persistence regulation.

3. Both PPP2R1A and NHSL1 were associated with RAC1 Q61L. Do they bind to the latter in the form of a complex or in a competitive manner? Was the interaction between PPP2R1A and NHSL1 enhanced or inhibited when RAC1 was activated? It's unclear how the association between PPP2R1A and WSC was modulated by signaling.

4. Does PPP2R1A affect the phosphorylation of RAC1?

5. For figure S5d (right panel), the labels are not consistent with the color of the lines. What is the red line? Also there's no orange-colored line in the figure.

6. The authors proposed that PPP2R1A were associated with NHSL1 fragment 4 with HEAT1 domain. But the mutations are in HEAT5 or HEAT7. How do these mutations change the structure of PPP2R1A and affect its binding to NHSL1?

7. In the last part of the paper, the authors investigated three inactive mutants of PPP2R1A, which happen most frequently in endometrial cancers. However, this work is mostly done on breast epithelial cells and breast cancer cells in some cases. The authors may discuss how these mutations are relevant to breast cancer.

8. PPP2R1A is considered as a tumor suppressor mostly because the phosphatase activity of PP2A counteracts with the effects of many oncogenic kinases. However, the function of PPP2R1A to positively regulate migration persistence reported here seems to support a pro-cancer progression role rather than a tumor suppressor, since cell migration is required for cancer metastasis. A more in-depth discussion of the results should be given in light with literature especially in the context of cancer metastasis. Besides, the potential role of PPP2R1A in regulating metastasis should be evaluated in a mouse model before drawing any conclusions.

Reviewer #2 (Remarks to the Author):

The present manuscript identifies PPP2R1A as an interaction of ABI1 and proposes that PPP2R1A is associated with NHSL1, which forms an alternative WAVE Shell Complex (WSC) devoid of WAVE and capable of regulating migration persistence.

The works contain some potentially novel findings, but the model that is proposed is not sufficiently supported by a cogent set of evidence and significantly more work (employing, for example, basic biochemical reconstitution experiments) appears needed (Below is a set of specific points that illustrate examples of what could be done). It must also be said the newly coined WAVE Shell Complex (WSC) is really a misnomer as it should not contain WAVE proteins at all.

Specific points

Figure 1e describes an increase in migratory persistence brought about by the expression of RAC1 Q61L. This is in line with the provided Movies 2 and with similar, previously-published observations by the same authors. However, in various cell lines the ectopic expression of constitutively active RAC1 leads to the aberrant formation of isotropically extended lamellipodia that end up impairing effective directional migration. Can the authors clarify this point and the different nature of their findings? Is this because

the cell lines employed expressed RAC1 Q61L in heterozygosity, and thus presumably at low levels? The levels of expression of RAC1 Q61L should be shown.

Within the same set of data, it is also shown that the ablation of PPP2R1A reduces migration persistence in a process that is rescued by the expression of RAC1 Q61L, suggesting that RAC1 activation might be downstream of PPP2R1A. Is the migration persistence increase due to ectopic expression of PPP2R1A dependent on RAC1 activity?

In Figure 2D, the authors produce a set of TAP experiments that lead them to conclude that "NHSL1 is the subunit of this alternative (WRC) complex that is recognized by PPP2R1A, and does not contain WAVE proteins". To corroborate this critical notion and central findings of the manuscript, however, more experiments are needed. One obvious prediction of the model proposed is that NHSL1 should act by competitively interacting with WAVE protein for the association with the ABI1/CYF1P1/NCKAP1/BRK1 complex. The authors are in an ideal position to provide such evidence given their established ability to produce and purify the wave complex in vitro.

In addition, a possible outcome of ectopically expressing PPP2R1A in the presence of NHSL1 would be the formation or sequestration of most of the WRC complex components, with the exception of WAVE molecules.

This suggests that these free WAVE proteins might be more readily degraded, and this should ultimately affect actin polymerization and lamellipodia formation. None of this, however, has been tested.

Also, as pointed out by the authors, NHSL1 was shown to interact with the whole WRC complex via an association with the ABI1 SH3 domain. This appears inconsistent with the present set of findings. Indeed the authors stated that: "NHSL1 was recently reported to interact with the complete canonical WRC through the ABI1 SH3 domain. PPP2R1A172 thus does not bind to all pools of NHSL1 molecules, but appears to select a specific conformation of NHSL1, where NHSL1 fully replaces WAVE within its complex". This statement while describing a plausible scenario, is, however, unsupported by the data presented and it remains unclear whether the two pools of NHSL1 exist at all. Again biochemical experiments with ideally purified molecules would seem needed to support the existence of different NHSL1-containing complexes. What would be the role if any of a WRC complex containing NHSL1 (that interacts via its proline-rich region with ABI1)?

In Figure 2F, alpha fold2 modeling is used to provide evidence that NHSL1 can interact with CYF1P1, NCKAP1, BRK1, and ABI2. There are predictions emerging from this modeling that should experimentally be tested: For example, the model predicts that 3 hydrophobic residues of NHSL1, upstream of its WHD, interact with CYFIP. What is the effect of mutants in these residues?

Finally, Finally, NHSL1 is shown and proposed to contain an N-terminal WAVE homology domain enabling its docking into the WAVE-less WRC complex. This complex should be devoid of the ability to activate ARP2/3 mediated actin polymerization. How than the interaction with PPP2R1A is required for RAC1 induce actin polymerization remains unclear. In the model proposed in Figure 6F, the coupling of PPP2R1A with NHSL1 and the WSC is proposed to act in promoting direction persistence, but how this might occurs remains at this stage unclear, particularly since the WSC should be incapable of promoting actin polymerization.

Reviewer #3 (Remarks to the Author):

The authors of the manuscript „PPP2R1A Regulates Migration Persistence through the WAVE Shell Complex” describe several new findings:

Firstly, they show that NHS and its paralog NHSL1 can replace WAVE (the activity bearing subunit) in the WAVE-regulatory complex (WRC) thereby building a complex that contains all WRC subunits except WAVE and thus being incapable of Arp2/3 activation. They tentatively name this complex WAVE-Shell Complex (WSC)

Secondly, they find that the regulatory subunit PPP2R1A of the PP2A phosphatase complex is specifically associated with WSC and not WRC

Finally they find that these two new players in WRC-regulation have opposite effects on cell migrational persistence but that these effects depend on each other.

The experiments presented here are of very good quality and convincingly support these findings. However, these are still many open ends and the interpretations – in my opinion – go too far and are not fully supported by the data.

Together, this reviewer really liked the biochemistry and the multitude of assays used to build the story. However, the conclusions should be sharpened and some additional experiments might help to support the point thy want to make. Moreover, the text often focusses on PPP2R1A although its role and mode of action remains elusive. On the other hand, the discovery and description of WSC is a significant step in the understanding of WRC mediated lamellipodia, which can be emphasized.

Specific comments:

The scheme in Figure 1c is misleading because it depicts and PPP2R1A interaction with WRC, although the authors convincingly show, that PPP2R1A interacts with WSC and not with WRC. There is no need to confuse the readers by depicting what the authors used to believe in the beginning of their research. They start with describing the interaction partners of Abi, which was already described to act outside of WRC, with IPs followed by mass spectrometry. The experiment show in Figs 1 a,b shows that the interaction of Abi-protein with PPP2R1A decreases when Rac signaling is high (WRC input signal) or when the output namely Arp2/3 complex is inhibited. Both conditions have in common that activating signals accumulate on WRC. However, the experiment also shows that the reduced interaction is not observed when both Rac is active AND Arp2/3 is inhibited at the same time. This is the first indication for PPP2R1A not directly associating with WRC.

In the following migration assays, Wang et al show that PPP2R1A has a positive effect on cell migration, which is abolished in the presence of constitutively active Rac, when the interaction with Abi is low. The authors next probe the interaction pattern by now pulling on PPP2R1A.

Here they make the striking finding that PPP2R1A exclusively interacts with Cyfip, NckAP1, Abi and Brck but not WAVE. Moreover, they show that the known negative regulator of cell migration NHSL1 is present in the IPs. In a tandem affinity approach, they then show that these components form a stable complex with NHSL1 replacing Wave. Finally, they clearly state and show that this complex (tentatively termed WSC) is the binding partner of PPP2R1A and not WRC as they fail to ever detect WAVE protein in PPP2R1A pulldowns.

It must be clear (and I missed that clarity in the text) that WSC lacks the capability to activate Arp2/3 complex and thus to induce actin assembly. Moreover, it is known from previous work that NHSL1 has a negative effect on cell migration. Therefore, the positive effect of PPP2R1A on cell migration must come from a suppression of the negative NHSL1- (or better WSC-) effect. So, PPP2R1A appears to act as a suppressor of the inhibitor (WSC) of migration.

The authors do not draw this conclusion although it appears to be the most likely explanation for most of their data.

However, there are less likely alternative explanations: previous reports indicate a close connection and positive correlation between PP2A activity and Rac activity (e.g. Wang et al 2014, Kowluru 2020, Ke et al., 2013). Therefore, siRNA towards PPP2R1A might lead to reduced Rac activity and overexpression of PPP2R1A might induce increased Rac activity in cells and explain the effects on migrational persistence. This connection would also explain why constitutively active Rac cancels the effect of PPP2R1A on migrational persistence. A potential effect of PPP2R1A RNAi or overexpression on endogenous Rac activation levels could be easily measured by Rac activation assays such as GLISA.

The interaction between PPP2R1A and WSC seems to be quite weak (Fig 2c). To what extent is endogenous NHSL1 engaged in WSC? Is there free NHSL1? How much percent of e.g. Cyfip is at a given point in time engaged with WRC or WSC? It would be very helpful to learn how stable WSC is and if WRC and WSC can interchange by changing WAVE for NHSL1. The fact that free WAVE was never detected can be explained by rapid degradation of uncomplexed WAVE but what about NHSL1. Is there a free pool? In other words: Do we deal with two stable complexes that compete for a seat in the first row (lam tip) upon Rac activation, one being capable of nucleating actin (WRC) and the other not (WSC)? Or do we have one stable core complex, normally containing WAVE but under certain conditions

(depending on PPP2R1A?) ripped for WAVE and substituted with NHSL1? This would be conclusive with the finding that PPP2R1A is not compulsory but depends on NHS1L.

Is it possible to imagine an assay in which this potential exchange can be shown? Can one immobilize WRC on beads and then incubate with WT or PPP2R1A-KO cell extracts followed by probing if NHSL1 then shows up on the beads dependent on the presence of PPP2R1A?

The bead assay with Rac-L61 is not very conclusive as it suggests a huge role for PPP2R1A which is not reflected by the other assays. According to the bead assay PPP2R1A would be almost essential for WRC mediated actin assembly although all other assays show that it is just a modulator.

Is it possible to test which proteins are recruited to the beads during that assays?

PPP2R1A RNA interference has a dramatic effect despite the presence of constitutively active L61 Rac on the beads, which abrogates/redundantizes the effects of PPP2R1A according to the migration assays and Abi pull downs. How can this be reconciled?

The sentence in lines 348-50 is also not helpful in this respect. ("The in vitro reconstitution assay of RAC1- dependent branched actin structures we report here using cell extracts is likely to play an important role in testing different scenarios for a multistep WAVE activation cycle") Why bother to show an inconclusive assay here when it is only becomes important much later when we know more?

Phosphorylation studies: It was a brave of the authors to attempt finding changes in the phospho-proteome of lamellipodial components. However, it is not so surprising that they did not find any. As the authors point out correctly, PPP2R1A might either play a scaffolding role without its catalytic subunit or the authors simply did not find the important (de) phosphorylated residue. The answer might even be Rac itself, which can be inhibited by phosphorylation on Ser71 (Kwon et al., 2000). This modification can be probed by Western blotting. Dephosphorylation via PP2A would explain the Rac activation-effect described earlier (and see above). The text part on WAVE Ser308 should be toned down or removed as this is a very unlikely candidate (PPP2R1A interacts with WSC not WRC).

The localization studies: The images in Fig 3 actually show that there is no co-localization between PPP2R1A and NHSL1. Whereas NHSL1 clearly is a tip component PPP2R1A is not. It even does not seem to fully recapitulate the actin-Arp2/3 complex-gradient. The authors need to show by e.g. FRAP assays, if the turnover of the two proteins at the lam tip speaks for interaction and that on the other hand the portion of PPP2R1A further back in the lamellipod is turned over at other rates than PPP2R1A at the lam tip. Is PPP2R1A turnover similar to that of NHSL1 in the places of putative colocalization?

We thank the reviewers for the considerable time and effort spent in carefully analyzing our dense manuscript. The comments of each reviewer led to significant improvements. Now that the revision process is finished, we are very pleased to have performed the requested experiments and to have achieved better presentation of the results.

Reviewer #1

In this paper, the authors identified PPP2R1A to be differentially associated with the WAVE complex subunit ABI1 when RAC1 was activated and downstream generation of branched actin was blocked. PPP2R1A was found to associate with the WAVE Shell Complex (WSC), that contains NHSL1 instead of WAVE proteins as in the canonical WAVE Regulatory Complex (WRC). The authors showed that PPP2R1A was required for cell migration persistence and RAC1-dependent actin polymerization via interaction with NHSL1. In addition, cancer-associated PPP2R1A mutations impaired its WSC binding and migration regulation. The findings provide information to better understand the molecular mechanism by which actin polymerization is modulated. However, some necessary information is still missing, especially the biological consequences of the binding of PPP2R1A to WSC and how such interaction was regulated by signaling.

Introduction

1. In the second paragraph, the authors may consider to provide more comprehensive information about the components of the WRC complex.

We have followed the advice of the reviewer in expanding the information about the WRC (composition and function) in the revised version of the manuscript (2nd paragraph of the introduction).

2. I would suggest to provide more information regarding PP2A and PPP2R1A in the introduction section.

As suggested, we have added the most important information concerning PPP2R1A and PP2A in the last paragraph of the introduction section, where we summarize the main findings of the manuscript. It was not possible to develop this part further in the introduction, since there was no connection between PPP2R1A and the WRC known before our work.

Results

For the label-free proteomics study,

3. The authors may want to provide more details regarding the data analysis and processing procedure. For example, how was data normalization performed for quantification?

We now provide detailed information for mass spectrometry methods, a section which is in the Supplementary Information. The relevant part reads:

Quantification was performed in label-free LFQ normalization mode ² using at least 2 razor or unique peptide per protein. Quantities were estimated using LFQ intensities and normalized by the intensity of the bait protein, ABI1 in the case of Table S1. Proteins found in control samples were filtered out as described in the tables. Significant changes in protein amounts were estimated by ANOVA with Bonferroni's Post-Hoc test using a p-value cutoff of 0.05.

4. The authors identified 89 proteins as ABI1 binding partners. However, it's unclear how they were filtered out. Were all proteins observed in the FLAG-GFP samples regarded as background interference? Or was quantitative comparison performed to remove background proteins? More details should be provided in the method section. In addition, many keratin isoforms were listed as ABI1 binding proteins in Table S1. Keratins are normally considered as contaminants in coIP-MS experiments.

We have now filtered out proteins that were also present in controls. This allowed us to get rid of keratins. In the case of Table S1, we nonetheless kept proteins that were detected in FLAG-GFP controls with a single peptide, otherwise we would even lose the bait itself, ABI1. We made a single exception to this rule for NCKAP1 that is a WRC subunit, and for which up to 3 peptides were detected in controls. This method of filtering allowed us to keep only 43 proteins, as specific interactors of ABI1.

5. More information should be provided in Table S1 such as protein accession number, detection coverage, and number of identified peptides. The information of the control samples (FLAG-GFP) should be included. Additionally, the quantification results should be provided for all identified proteins together with their P values from statistical analysis.

We have now included the requested information in Table S1 and also in the other tables summarizing mass spectrometry results.

6. For figure 1b, it would be easier to understand the results by showing the fold change of treatment/control instead of the other way around. For example, MCF10A+CK666/MCF10A and RAC1 Q61L+CK666/RAC1 Q61L. In addition, I would suggest to present the results in column graphs or box plots with the significance levels indicated in the figure. All the detailed information about the quantitative analysis could be provided in Table S1.

The reviewer is right that the previous table was hard to read. We now graphically display in Fig.1b the variations with a bar chart plotting the log₁₀ of various ratios, where the treated condition is systematically the numerator and the control condition in the denominator. We used the log₁₀ of the ratio to give the same weight to up- and down-regulations. We thank the reviewer for this suggestion, which greatly improved the presentation of these variations.

Of note, the important variations of PPP2R1A levels in the 3 biological repeats did not reach statistical significance, whereas small variations of WRC subunits for example did. For this

reason, we decided not to display p-values in the plot of Fig.1b, but all p-values can be found in Table S1.

7. It seems that different modulators responded quite differently to RAC1 activation and Arp2/3 inhibition. I think the authors should discuss about their observations in more details. For example, the authors mentioned that it was consistent with the literature that more lamellipodin were associated with WRC when RAC1 was activated. Based on the results, such interaction was also enhanced when MCF10A cells were treated with CK666. Is this expected and supported by existing knowledge? Similar to PPP2R1A, lamellipodin also responded differently to CK666 treatment in parental and RAC1 Q61L cells. The authors mentioned that RAC1 Q61L did not depend on the feedback of branched actin for its activity. But it's still unclear to me why CK666 treatment affected the association of PPP2R1A to ABI1 in RAC1 Q61L cells.

We fully agree with the reviewer that the comparison of the two cell lines, parental and RAC1 Q61L expressing cells, does not yield a clear idea of the mechanism. For example, as noted by the reviewer, CK-666 treatment of parental cells decreases the amount of PPP2R1A associated with ABI1, whereas the same treatment of RAC1 Q61L expressing cells increases this association. The same holds true for Lamellipodin, although to a lesser extent. The mention of the RAC1 feedback loop in this context does not allow a better understanding of these data and so we removed this interpretation.

The only potential explanation we can come up with to understand the variations displayed in Fig.1b is the fact that we compare two different cell lines. It means that cells had time to adjust their physiology to the continuous expression of RAC1 Q61L, by regulating, for example, expression of genes mediating the feedback. Since we do not know the exact causes of these different responses to CK-666, we did not emphasize further the detailed description of these differential results, but rather use the main text to indicate that these differential responses allow us to pinpoint PPP2R1A as a good candidate to regulate migration persistence among many other proteins. The rest of the manuscript validates the candidate.

8. For figure 1c, the model is a little misleading. It shows the dynamic interaction between PPP2R1A and WRC. However, the authors later showed that PPP2R1A bound to WSC.

We agree with this comment. This point was also raised by another reviewer (point #1 of reviewer #3). We have removed this unnecessary and temporary working model from figure 1 to avoid confusion.

For the other parts,

9. The authors treated the cells with CK666 to inhibit Arp2/3. What concentration was used? For how long? Did the author test the effects of different doses? The authors should show how the treatment affect the activity of Arp2/3. How does this treatment affect RAC1?

The reviewer is right that the concentration and duration of treatment should be indicated in figure legends. This oversight has now been corrected. We now mention in the legend of figure 1 that CK-666 was used at 100 μ M for 16 h, the total duration of our migration assays. This treatment was shown to block migration persistence of both MCF10A and the genome-edited MCF10A cell line expressing RAC1 Q61L (Fig.S8 in Molinie et al, Cell Res 2019). This concentration is above the IC50 of CK-666 that we characterized in details for Arp2/3-mediated actin polymerization in vitro, lamellipodium protrusion and cell migration, in a recent publication that compares the effect of CK-666 to novel analogs we identified (Fokin et al., Frontiers Pharmacol 2022). As requested, we also evaluated the effect of this treatment with CK-666 on RAC1 activation by an ELISA assay and found that it has no effect. These data have been added in the new supplementary figure 1 (Fig.S1b).

10. The authors tried to study phosphorylation alteration of the WSC or WRC by TAP purification of ABI1 and MS analysis. In my opinion, this is not a very effective strategy. First, the abundance of proteins obtained by TAP purification is normally low. Second, large experimental variation is expected, therefore small quantitative changes cannot be efficiently detected. Third, phosphopeptides normally exist at much lower levels compared to non-phosphopeptides, therefore difficult to be detected without enrichment. I'm surprised that PPP2R1A bound to WSC in the form of PP2A complex without affecting its phosphorylation. Then, what are the biological consequences of such binding? Not detecting any change does not necessarily mean its phosphatase activity is not involved. The author may consider to perform a large scale quantitative phosphoproteomics analysis with proper enrichment and quantification to seek more clues. I am also wondering whether inhibiting the phosphatase activity of PP2A would affect the functions of PPP2R1A in the context of migration persistence regulation.

The reviewer is obviously experienced in mass spectrometry of phosphosites and we agree with him or her that the lack of identification of phosphosites that depend on PPP2R1A does not rule out a role of the PP2A phosphatase activity. However, we did not follow the suggested experimental approach. During the revision, we found in the bead assay that PPP2R1A-mediated actin polymerization in cell extracts is not affected by treatment with 3 different PP2A phosphatase inhibitors. This important result renders the identification of phosphosites less important. Moreover, a global approach of phosphopeptide characterization when PPP2R1A is present or not has already been reported in the literature (Kauko et al., JBC 2020) and no differential phosphosite was reported on the WSC. Available evidence suggests that the PP2A phosphatase activity is unlikely to be involved in the here reported regulation of migration persistence. The arguments are now recapitulated in the first paragraph of the discussion section.

11. Both PPP2R1A and NHSL1 were associated with RAC1 Q61L. Do they bind to the latter in the form of a complex or in a competitive manner? Was the interaction between PPP2R1A and NHSL1 enhanced or inhibited when RAC1 was activated? It's unclear how the association between PPP2R1A and WSC was modulated by signaling.

We were showing in Fig.S4b (and now renumbered Fig.S7b of the revised version) that RAC1 Q61L binds to NHSL1 and PPP2R1A to a lower extent. There is no straightforward and convincing way to answer the question, since the WSC harbours at least three described binding sites for active RAC1, one on NHSL1 (Law et al, Nat Commun 2021) and two on CYFIP1 which is a subunit shared by WRC and WSC (Chen et al., eLife 2017). PPP2R1A, which is less enriched than the WSC in the GST-RAC1 Q61L pull-down, might simply be dragged along.

This question of signaling is thus very complex, but we agree with the reviewer that it is an important question. When we checked how the interaction of PPP2R1A with the WSC was regulated, the only clear effect we found was that starvation of growth factors down-regulated both NHSL1 and PPP2R1A and as a consequence down-regulated the amount of WSC bound to PPP2R1A. These new data have been included in a new supplementary figure (Fig.S4a).

12. Does PPP2R1A affect the phosphorylation of RAC1?

RAC1 phosphorylation on serine 71 was indeed previously reported to regulate its interaction with the WRC (Schwarz et al, PLoS One 2012). This point was also raised by another reviewer (point #9 of reviewer #3). We specifically checked phosphorylation of RAC1 using the phosphospecific antibody of serine 71 and found no difference in phosphorylation levels when PPP2R1A was knocked-down, knocked-out or overexpressed. These new data have been included in figure S6 devoted to phosphorylation analyses.

13. For figure S5d (right panel), the labels are not consistent with the color of the lines. What is the red line? Also there's no orange-colored line in the figure.

We thank the reviewer for noting this error. We have corrected the panel previously displayed in Fig.S5d, now renumbered Fig.S9b.

14. The authors proposed that PPP2R1A were associated with NHSL1 fragment 4 with HEAT1 domain. But the mutations are in HEAT5 or HEAT7. How do these mutations change the structure of PPP2R1A and affect its binding to NHSL1?

We thank the reviewer for this comment, which shows how careful his or her reading was. Indeed PPP2R1A directly associates with NHSL1 through HEAT repeat #1 (Motifs 1 and 2). But we also report motif 3, which mediates the indirect association of PPP2R1A with NHSL1 through B regulatory subunits of the PP2A complex. We initially provided a structural model of NHSL1 motif 3 bound to PPP2R5D. During the revision, we found in the TAP of the WSC another B subunit, PPP2R5E, that similarly bridges PPP2R1A and NHSL1 motif 3. Revised figure 8 now displays the structural models of the two B subunits.

Since tumor associated mutations in HEAT repeats #5 and #7 affects binding of PPP2R1A to regulatory subunits (Haesen et al., Cancer Res 2016; Taylor et al., Cancer Res 2019), it suggests that the contribution of indirect binding of PPP2R1A to NHSL1 through these B regulatory subunits is critical. Motif 3 can create an avidity effect together with the motifs 1 and 2 which are involved in direct binding. This point is now discussed in the 2nd paragraph of the discussion section.

15. In the last part of the paper, the authors investigated three inactive mutants of PPP2R1A, which happen most frequently in endometrial cancers. However, this work is mostly done on breast epithelial cells and breast cancer cells in some cases. The authors may discuss how these mutations are relevant to breast cancer.

The reviewer is right that the PPP2R1A mutations we study here are most frequent in endometrial cancers. Nonetheless the recurrent mutation R183W is also reported in as many as 7 % of breast carcinomas (O'Connor et al, Oncogene 2020). This point is now made clear in the second paragraph of the discussion. Our study reports the function of PPP2R1A in regulating migration persistence of normal cells. MCF10A are mammary epithelial cells that are immortalized, but not transformed. Even if our paper is not a cancer study, we believe that it is worth reporting that recurring mutations of PPP2R1A found in tumors impair the control of migration persistence, the novel function of PPP2R1A we report here.

16. PPP2R1A is considered as a tumor suppressor mostly because the phosphatase activity of PP2A counteracts with the effects of many oncogenic kinases. However, the function of PPP2R1A to positively regulate migration persistence reported here seems to support a pro-cancer progression role rather than a tumor suppressor, since cell migration is required for cancer metastasis. A more in-depth discussion of the results should be given in light with literature especially in the context of cancer metastasis. Besides, the potential role of PPP2R1A in regulating metastasis should be evaluated in a mouse model before drawing any conclusions.

The reviewer is absolutely right that a positive role of PPP2R1A in migration persistence does not account for its tumor suppressor role. In the first version, we reported in a supplementary figure defective epithelial morphogenesis of acini developed by MCF10A when PPP2R1A is inactivated by KO or by tumor mutations. We have now included these data in the main figures (Fig.9c), because this implication in epithelial cell polarity is more likely to contribute to cancer progression than the positive role of PPP2R1A in migration persistence. We now discuss this important point in the second paragraph of the discussion section.

Positive feedback loops can sustain a behavior over time (here migration persistence) but also commit cells into a new differentiated state (here epithelial cell polarity). So in a way, it is not so surprising that mutations impairing migration persistence also impair cell polarity, since they might share the same mechanism. This speculation is just for the sake of sharing our views with the reviewer and was not included in the discussion to keep the manuscript simple.

We thank the reviewer for his or her careful analysis of our results and constructive criticism. The main improvements are the following:

- Better presentation of MS data (Fig1, Tables, Methods)
- Levels of WSC controlled by growth factors (starvation/replenishment experiment)
- Richer introduction and discussion sections

Reviewer #2

The present manuscript identifies PPP2R1A as an interaction of ABI1 and proposes that PPP2R1A is associated with NHSL1, which forms an alternative WAVE Shell Complex (WSC) devoid of WAVE and capable of regulating migration persistence.

The works contain some potentially novel findings, but the model that is proposed is not sufficiently supported by a cogent set of evidence and significantly more work (employing, for example, basic biochemical reconstitution experiments) appears needed (Below is a set of specific points that illustrate examples of what could be done). It must also be said the newly coined WAVE Shell Complex (WSC) is really a misnomer as it should not contain WAVE proteins at all.

The reviewer is right that the novel complex we identify and purify here does not contain any WAVE molecule, the only Arp2/3 activating subunit of the WAVE Regulatory Complex. We believe that the “WAVE Shell” conveys well that the fact that this complex is almost the WRC with no WAVE. At the same time, we agree that to define a new object negatively by the absence of WAVE is not fully satisfying. Therefore we now call this new complex the NHSL1-containing WAVE Shell Complex, including in the title of the manuscript. We keep the simple abbreviation of WSC for fluidity in the text. Calling this complex the NHSL1 complex would be ambiguous as well, since NHSL1 can bind to the WRC containing WAVE. Moreover, this nomenclature has the advantage to be potentially extended to other NHS family proteins, which also contain WAVE homology domains, in case other NHS-containing WSCs are found in the future, as discussed in the revised discussion (3rd paragraph).

During the revision, we have followed the reviewer’s recommendation to reconstitute the NHSL1-containing WAVE Shell Complex.

Specific points

1. Figure 1e describes an increase in migratory persistence brought about by the expression of RAC1 Q61L. This is in line with the provided Movies 2 and with similar, previously-published observations by the same authors. However, in various cell lines the ectopic expression of constitutively active RAC1 leads to the aberrant formation of isotropically extended lamellipodia that end up impairing effective directional migration. Can the authors clarify this point and the different nature of their findings? Is this because the cell lines employed expressed RAC1 Q61L in heterozygosity, and thus presumably at low levels? The levels of expression of RAC1 Q61L should be shown.

Several publications indeed reported that the transfection of RAC1 Q61L induces isotropic lamellipodial extension, thus preventing cell migration in any direction. The logical conclusion that can be drawn from these observations would be that RAC1 is a negative regulator of cell migration. We believe that everyone in the field agrees that the established RAC1 function is quite the opposite. As noted by the reviewer, isotropic lamellipodial extension and abolishment of cell migration are not what we observe here and in our previous publication, Molinie et al,

Cell Res 2019, using the genome-edited MCF10A cell line where the Q61L mutation is introduced into one of the *RAC1* alleles. We observed that migration of these genome-edited MCF10A cells is more persistent than that of parental cells. Our system is thus more physiological, since it is in line with the numerous publications that have established that *RAC1* is required for cell migration.

As requested, we have characterized *RAC1* levels and activity in the *RAC1* Q61L expressing cell line by the GST-PAK pull-down. As expected, *RAC1* is not overexpressed in the genome-edited cell line, but overactivated. These new data have been added in a new supplementary figure devoted to *RAC1* activity and levels (Fig.S1a).

2. Within the same set of data, it is also shown that the ablation of *PPP2R1A* reduces migration persistence in a process that is rescued by the expression of *RAC1* Q61L, suggesting that *RAC1* activation might be downstream of *PPP2R1A*. Is the migration persistence increase due to ectopic expression of *PPP2R1A* dependent on *RAC1* activity?

We agree with the reviewer that the rescue of migration persistence of *PPP2R1A* knock-down by *RAC1* Q61L is compatible with the idea that *RAC1* is downstream of *PPP2R1A*. Nonetheless, in the presence of positive feedback loops, such as the ones that control migration persistence, reviewed in Krause & Gautreau, Nat Rev Mol Cell Biol 2014, it is difficult to distinguish a downstream effector from an upstream activator.

We have measured *RAC1* activation in the stable MCF10A cell line that overexpresses FLAG-GFP-*PPP2R1A* using GST-PAK pull-down and a commercial ELISA kit (G-LISA from Cytoskeleton, Inc). With both methods, we found no overactivation of *RAC1* induced by *PPP2R1A* overexpression. These data are included in the new Supplementary Figure 1 (Fig.S1c, S1d, S1e). An overactivation would be expected if *PPP2R1A* was upstream of *RAC1*.

To know whether *RAC1* is required for the increased migration persistence in the stable *PPP2R1A*-overexpressing MCF10A cell line, we inactivated *RAC1* in both *PPP2R1A*-overexpressing and parental MCF10A cells using two independent methods, the NSC23766 chemical inhibitor that blocks *RAC1* activation and siRNA-mediated depletion of *RAC1*. The results are ambiguous and are displayed on the next page for the reviewer only (**Fig.R1**).

The chemical inhibitor had surprisingly no effect on migration persistence in both parental and *RAC1* Q61L expressing cells. We used it at the highest possible concentration, 25 μ M. At concentrations higher than 25 μ M, it shows toxicity. The same chemical inhibitor was found during the revision to block *RAC1*-induced actin polymerization in vitro using our bead assay and MCF10A cell extracts.

Figure R1. Role of RAC1 in MCF10A cells expressing FLAG-GFP-PPP2R1A. (a) FLAG-GFP-PPP2R1A or FLAG-GFP expressing cells were treated for 16 h with 25 μ M of the Rac1 inhibitor NSC23766 and their migration as single cells was studied during the treatment. Tracking 7 h, n=22. (b) FLAG-GFP-PPP2R1A or FLAG-GFP expressing cells were transfected with the two siRNAs against Rac1 for 48 h and their migration as single cells was studied. Tracking 7 h, n=40. Cell trajectories, migration persistence, speed and MSD are displayed. 3 biological repeats with similar results, only one is displayed. *P<0.05; ** P<0.01; n.s. not significant.

In contrast, RAC1 depletion using two siRNAs increased migration persistence in both parental and RAC1 Q61L expressing cells. This result may sound counter-intuitive, given that RAC1 is generally agreed to be required for cell migration (as discussed in the point 1, just above). Such an observation has, however, been reported previously by the group of Kenneth Yamada. A decrease in RAC1 levels renders cells more directionally persistent (Pankov et al., JCB 2005). Their interpretation is that cells with decreased RAC1 levels are less likely to form lateral lamellipodia that render cell migration more random. These data about RAC1 requirement in migration persistence are not even included as supplementary data, since they are likely to confuse readers more than providing explanations accounting for the main results of the manuscript. The ambiguity stems from the realization that migration persistence can be impaired by too much or too little RAC1 activity (Pankov et al. JCB 2005).

3. In Figure 2D, the authors produce a set of TAP experiments that lead them to conclude that "NHSL1 is the subunit of this alternative (WRC) complex that is recognized by PPP2R1A, and does not contain WAVE proteins". To corroborate this critical notion and central findings of the manuscript, however, more experiments are needed. One obvious prediction of the model proposed is that NHSL1 should act by competitively interacting with WAVE protein for the association with the ABI1/CYFIP1/NCKAP1/BRK1 complex. The authors are in an ideal position to provide such evidence given their established ability to produce and purify the wave complex *in vitro*.

There are in fact two points in this comment of the reviewer. The first one refers to the coexistence of two distinct complexes, the WRC containing WAVE and the WSC containing NHSL1. Since PPP2R1A TAP retrieves all common subunits and NHSL1, but not WAVE, it is logical to think that PPP2R1A specifically interacts with the WSC through NHSL1. This idea was experimentally tested. siRNA-mediated depletion of NHSL1 indeed prevented the interaction of PPP2R1A with common subunits (Fig.2d). These data are clear and simple provided that one is convinced by the existence of the two alternative complexes. We had purified the WSC. However, the reviewer advised to reconstitute the WSC to further prove its existence.

We used *in vitro* translation of subunits to reconstitute the WSC, as we previously did for the WRC (Gautreau et al., PNAS 2004). This allowed us to detect that the WHD of NHSL1 can indeed substitute for WAVE2 in the formation of a pentameric complex with ABI1/CYFIP1/NCKAP1/BRK1 (Fig.3a). When the two proteins are coexpressed, they indeed compete for the interaction with common subunits (Fig.3b). We went further by collaborating with the group of Baoyu Chen from the University of Iowa, who had reconstituted the WRC from recombinant proteins. Three new authors are associated with the revised manuscript, because his group managed to reconstitute the WSC from pure recombinant proteins (Fig.3c). The WSC behaved like the WRC in gel filtration (Fig.3d). The WSC was found by electron microscopy after negative staining to have a size and shape very similar to the ones of the WRC (Fig.3e). We thank the reviewer for this suggestion, because the reconstitution after the purification alleviates all possible doubts concerning the possibility and the existence of the WSC.

4. In addition, a possible outcome of ectopically expressing PPP2R1A in the presence of NHSL1 would be the formation or sequestration of most of the WRC complex components, with the exception of WAVE molecules. This suggests that these free WAVE proteins might be more readily degraded, and this should ultimately affect actin polymerization and lamellipodia formation. None of this, however, has been tested.

We tested this possibility using the stable MCF10A cell line that overexpresses PPP2R1A. We found that this cell line has the same amount of WAVE2 and of WRC as control cells, so the possibility evoked by the reviewer does not seem to occur. These new data have been added to the Supplementary Figure 2 (Fig.S2f). The mechanism by which PPP2R1A regulates cell migration is probably quite elaborate and more regulated than a simple competition between NHSL1 and WAVE subunits.

5. Also, as pointed out by the authors, NHSL1 was shown to interact with the whole WRC complex via an association with the ABI1 SH3 domain. This appears inconsistent with the present set of findings. Indeed the authors stated that: "NHSL1 was recently reported to interact with the complete canonical WRC through the ABI1 SH3 domain. PPP2R1A thus does not bind to all pools of NHSL1 molecules, but appears to select a specific conformation of NHSL1, where NHSL1 fully replaces WAVE within its complex". This statement while describing a plausible scenario, is, however, unsupported by the data presented and it remains unclear whether the two pools of NHSL1 exist at all. Again biochemical experiments with ideally purified molecules would seem needed to support the existence of different NHSL1-containing complexes. What would be the role if any of a WRC complex containing NHSL1 (that interacts via its proline-rich region with ABI1)?

We first want to make the point that there is no inconsistency. Matthias Krause, the senior author of the recent study that reported that NHSL1 can interact with ABI1 SH3 domains, endorses our conclusions, since he is a co-author of the present manuscript. The here reported interaction through the NHSL1 WHD is simply in addition to the previously reported mode of NHSL1 binding to the WRC. To experimentally support the two modes of binding in our manuscript, we used *in vitro* translation. We have fully confirmed that the NHSL1 C-terminus that contains the two previously reported ABI1 SH3 binding sites interacts with the WAVE2-containing WRC, whereas the NHSL1 N-terminus that contains the WHD assembles the WSC. These new experimental data, which are very important indeed, have been included in the main figure 3 (Fig.3a). This point of the two NHSL1 binding modes are also discussed (3rd paragraph) in the revised discussion section.

The precise role of the other complex, WRC bound to NHSL1, is beyond the scope of our manuscript centered on the role of PPP2R1A and its interaction with the WSC. The two complexes, NHSL1 bound to the complete WRC and NHSL1 bound to the WSC, share all components, except WAVE, which is missing in the latter. As a consequence, there is no specific way to address the question of their respective role using cell biology experiments.

6. In Figure 2F, alpha fold2 modeling is used to provide evidence that NHSL1 can interact with CYF1P1, NCKAP1, BRK1, and ABI2. There are predictions emerging from this modeling that should experimentally be tested: For example, the model predict that 3 hydrophobic residues of NHSL1, upstream of its WHD, interact with CYFIP. what is the effect of mutants in these residues?

Fig.2f shows the overall structural model of the WSC produced by AlphaFold2. The 3 N-terminal residues of NHSL1, upstream of its WHD, that interacted with CYFIP1 were highlighted in the previous Figure S2. We were not claiming that they had an essential role in the original version of our manuscript. We were just showing them, because equivalent residues in WAVE were not known to participate in the WRC. During the revision, we found that these residues upstream of the WHD were not essential, because the WSC was reconstituted with a form of NHSL1 that does not contain these residues (Fig.3c, Fig.S3). So this supplementary figure focusing on residues upstream of the WHD was misleading. We replaced it with a new supplementary figure showing the heterotrimeric coiled coil that NHSL1 WHD forms together with BRK1 and ABI2. To experimentally address the importance of the coiled coil, we mutated the conserved hydrophobic F58 into a charged residue, D. The F58D mutation indeed impairs NHSL1 incorporation into the WSC (Fig.S3). The heterotrimeric coiled coil formed by the WHD is thus more important than residues upstream of the WHD and deserves this supplementary figure.

7. Finally, NHSL1 is shown and proposed to contain an N-terminal WAVE homology domain enabling its docking into the WAVE-less WRC complex. This complex should be devoid of the ability to activate ARP2/3 mediated actin polymerization. How than the interaction with PPP2R1A is required for RAC1 induce actin polymerization remains unclear. In the model proposed in Figure 6F, the coupling of PPP2R1A with NHSL1 and the WSC is proposed to act in promoting direction persistence, but how this might occur remains at this stage unclear, particularly since the WSC should be incapable of promoting actin polymerization.

We absolutely agree with the reviewer that the WSC should be devoid of the ability to activate Arp2/3-mediated actin polymerization. This is now clearly stated in the discussion section (4th paragraph). Like the reviewer, we would really like to know the exact role of the WSC, but we did not figure it so far. We believe that the exact role of the WSC is beyond the scope of the current paper, which is already particularly dense. We will address this important question in future studies. To know more about what has been attempted, we refer the reviewer to a detailed answer we made to the point 6 of reviewer #3.

It is useful at this point to recapitulate what our current work has established. We have identified a novel regulator of cell migration, PPP2R1A. We have carefully studied what PPP2R1A does – to regulate migration persistence – and in which condition – when RAC1 is not constitutively activated and when NHSL1 is present. We have also figured the mechanism by which PPP2R1A regulates migration persistence – through the coupling to the here uncovered alternative form of the WAVE complex, the NHSL1-containing WAVE shell complex (WSC), where WAVE is replaced by NHSL1. We have been able to purify and reconstitute the WSC.

We thank the reviewer for his or her careful analysis of our results and constructive criticism. The main improvements are the following:

- Reconstitution of the WSC by in vitro translation and competition between WAVE and NHSL1
- Confirmation that NHSL1 binds to the WRC through the C-terminus, whereas WAVE binds to the WSC through the N-terminal WHD, which forms an important trimeric coiled coil
- Reconstitution of the WSC using recombinant proteins and its characterization by gel filtration and EM
- Measurements of RAC1 levels and activity

Reviewer #3

The authors of the manuscript „PPP2R1A Regulates Migration Persistence through the WAVE Shell Complex” describe several new findings:

Firstly, they show that NHS and its paralog NHSL1 can replace WAVE (the activity bearing subunit) in the WAVE-regulatory complex (WRC) thereby building a complex that contains all WRC subunits except WAVE and thus being incapable of Arp2/3 activation. They tentatively name this complex WAVE-Shell Complex (WSC). Secondly, they find that the regulatory subunit PPP2R1A of the PP2A phosphatase complex is specifically associated with WSC and not WRC. Finally they find that these two new players in WRC-regulation have opposite effects on cell migrational persistence but that these effects depend on each other.

The experiments presented here are of very good quality and convincingly support these findings. However, there are still many open ends and the interpretations – in my opinion – go too far and are not fully supported by the data.

Together, this reviewer really liked the biochemistry and the multitude of assays used to build the story. However, the conclusions should be sharpened and some additional experiments might help to support the point they want to make. Moreover, the text often focusses on PPP2R1A although its role and mode of action remains elusive. On the other hand, the discovery and description of WSC is a significant step in the understanding of WRC mediated lamellipodia, which can be emphasized.

We thank the reviewer for the nice appreciation of our work and for helping us to better present the study.

As for our choice to focus the study on PPP2R1A, this was deliberate for several reasons. The first one is that PPP2R1A is the true original regulator of cell migration that our work uncovers. NHSL1 was recently reported in a thorough study by Matthias Krause (Law et al., Nat Commun 2021). The second reason is that without PPP2R1A, we would not have discovered the WSC, since NHSL1 binds to the WRC through the ABI1 SH3 domain (Law et al., Nat Commun 2021).

Nonetheless we have significantly strengthened the characterization of the WSC in the revised version, in particular through its reconstitution.

Specific comments:

1. The scheme in Figure 1c is misleading because it depicts and PPP2R1A interaction with WRC, although the authors convincingly show, that PPP2R1A interacts with WSC and not with WRC. There is no need to confuse the readers by depicting what the authors used to believe in the beginning of their research.

We are convinced by the argument. This point was raised by another reviewer (point #8 of reviewer #1). We have thus removed this scheme from the revised version.

2. They start with describing the interaction partners of Abi, which was already described to act outside of WRC, with IPs followed by mass spectrometry. The experiment show in Figs 1 a,b shows that the interaction of Abi-protein with PPP2R1A decreases when Rac signaling is high (WRC input signal) or when the output namely Arp2/3 complex is inhibited. Both conditions have in common that activating signals accumulate on WRC. However, the experiment also shows that the reduced interaction is not observed when both Rac is active AND Arp2/3 is inhibited at the same time. This is the first indication for PPP2R1A not directly associating with WRC.

The reviewer is right that logics of these facts are not obvious. This point was also raised by reviewer #1 (point #7). We refer to the detailed answer we already have provided. Briefly, RAC1 Q61L expressing cells have probably adjusted their physiology to the constitutive activation of RAC1 signaling. As for the point that PPP2R1A is not directly associated with WRC, indeed the rest of the manuscript proves that PPP2R1A associates instead with an alternative complex, the WSC. We also refer the reviewer to our answer of point #14 of reviewer #1 concerning the direct and indirect interaction of PPP2R1A with the WSC.

3. In the following migration assays, Wang et al show that PPP2R1A has a positive effect on cell migration, which is abolished in the presence of constitutively active Rac, when the interaction with Abi is low. The authors next probe the interaction pattern by now pulling on PPP2R1A.

Here they make the striking finding that PPP2R1A exclusively interacts with Cyfip, NckAP1, Abi and Brck but not WAVE. Moreover, they show that the known negative regulator of cell migration NHSL1 is present in the IPs. In a tandem affinity approach, they then show that these components form a stable complex with NHSL1 replacing Wave. Finally, they clearly state and show that this complex (tentatively termed WSC) is the binding partner of PPP2R1A and not WRC as they fail to ever detect WAVE protein in PPP2R1A pulldowns. It must be clear (and I missed that clarity in the text) that WSC lacks the capability to activate Arp2/3 complex and thus to induce actin assembly.

We absolutely agree with the reviewer and apologise for this oversight. This point was also raised by reviewer #2 (point #7). This is now clearly stated in the revised discussion (4th paragraph).

4 Moreover, it is known from previous work that NHSL1 has a negative effect on cell migration. Therefore, the positive effect of PPP2R1A on cell migration must come from a suppression of the negative NHSL1- (or better WSC-) effect. So, PPP2R1A appears to act as a suppressor of the inhibitor (WSC) of migration. The authors do not draw this conclusion although it appears to be the most likely explanation for most of their data.

We agree with the reviewer that it would be a simple explanation and thank him or her for the suggestion. This idea is now presented in the revised discussion (3rd paragraph).

5 However, there are less likely alternative explanations: previous reports indicate a close connection and positive correlation between PP2A activity and Rac activity (e.g. Wang et al 2014, Kowluru 2020, Ke et al., 2013). Therefore, siRNA towards PPP2R1A might lead to reduced Rac activity and overexpression of PPP2R1A might induce increased Rac activity in cells and explain the effects on migrational persistence. This connection would also explain why constitutively active Rac cancels the effect of PPP2R1A on migrational persistence. A potential effect of PPP2R1A RNAi or overexpression on endogenous Rac activation levels could be easily measured by Rac activation assays such as GLISA.

We fully agree with the reviewer that a positive correlation between PPP2R1A levels and RAC1 activation would account for the data. We measured RAC1 activation levels by GLISA, as suggested, and also by the PAK pull-down assay in order to see RAC1 levels by Western blot. We found RAC1 levels and activity were decreased in both PPP2R1A knock-down and knock-out. Results were sometimes at the limit of significance, but clearly point towards this direction. The converse experiment of PPP2R1A overexpression, however, did not show significant upregulation of RAC1 levels or activity. These new data are now presented in the new Supplementary Figure 1, together with the measurements of RAC1 activation requested by reviewer #1. We believe that these results, even though they do not provide a unified explanation for the effects on migration persistence, are interesting and contribute to the characterization of the many genetically-perturbed cell lines we have generated for this study.

6. The interaction between PPP2R1A and WSC seems to be quite weak (Fig 2c). To what extent is endogenous NHSL1 engaged in WSC? Is there free NHSL1? How much percent of e.g. Cyfip is at a given point in time engaged with WRC or WSC? It would be very helpful to learn how stable WSC is and if WRC and WSC can interchange by changing WAVE for NHSL1. The fact that free WAVE was never detected can be explained by rapid degradation of uncomplexed WAVE but what about NHSL1. Is there a free pool? In other words: Do we deal with two stable complexes that compete for a seat in the first row (lam tip) upon Rac activation, one being capable of nucleating actin (WRC) and the other not (WSC)? Or do we have one stable core complex, normally containing WAVE but under certain conditions (depending on PPP2R1A?)

ripped for WAVE and substituted with NHSL1? This would be conclusive with the finding that PPP2R1A is not compulsory but depends on NHS1L. Is it possible to imagine an assay in which this potential exchange can be shown? Can one immobilize WRC on beads and then incubate with WT or PPP2R1A-KO cell extracts followed by probing if NHSL1 then shows up on the beads dependent on the presence of PPP2R1A?

Several of the questions raised are not easily addressed, because the only way to be sure that NHSL1 is part of the WSC - and not simply associated with the whole WAVE containing WRC, as shown in Law et al. Nat Commun 2021 and in our new Fig.3a using *in vitro* translation - is to detect it when bound to PPP2R1A. But to obtain an idea of the different pools of these proteins, we performed fractionation of cell extracts by ultracentrifugation on sucrose gradients.

Protein distributions were compatible with the view that only the previously known complexes, the WRC and the PP2A complex were detected. NHSL1 peaked in fractions with an intermediate sedimentation coefficient compared to the ones of PP2A and WRC. We checked protein distributions in sucrose gradients upon PPP2R1A and NHSL1 knock-out. We obtained the new *NHSL1* KO MCF10A cell line for this purpose. In both cases, we did not detect any shift in the distribution of subunits shared between the WSC and other complexes, suggesting that the WSC does not constitute a major pool of its constituent subunits.

We thus decided to compare the abundance of WSC and WRC. To this end, we designed a TAP experiment where the first immunoprecipitation selects a common subunit, BRK1, and the two distinct complexes, WSC and WRC, are immunoprecipitated in a second step through GFP-PPP2R1A and endogenous WAVE2 respectively. This allowed us to estimate that the WSC is about 7 to 8 fold less abundant than the WRC. All these new data have been included in the new supplementary figures S4 and S5. The respective abundance of WSC and WRC was difficult to determine, but it was important to do it, and we thank the reviewer for this suggestion.

As for the potential exchange of subunits, this is very complicated to test on beads, because either we specifically immobilize the WRC through WAVE and then cannot detect the WSC if formed, or we immobilize a common subunit and then we already have the WSC in the initial situation.

We can share with the reviewer an interesting result. It is an old result that we did not publish. It is a follow-up of the reference Derivery et al. PLOS One 2008, where we electroporated the purified PC-tagged BRK1 homotrimer that is converted into a single molecule of BRK1 during WRC assembly. Here, we additionally metabolically labeled the electroporated cells using ³⁵S Met to detect newly synthesized subunits associated with the PC-tagged BRK1. The pool of neosynthesized subunits were found specifically associated with exogenous PC-BRK1, except neosynthesized WAVE2, which is found in the bulk of WRC. One possible interpretation for these data is that neosynthesized WAVE2, which is initially associated with PC-BRK1 like other WRC subunits, is able to escape the newly assembled WRC and re-associate with other WRCs. This shuffling of WAVE2 among different WRCs is unique to the WAVE subunits and might perhaps involve NHSL1-containing WSC as a way to maintain the shell when WAVE is away. Unfortunately, we were unable to test this hypothesis by performing RNAi of PPP2R1A or NHSL1, because stricter regulations in our new institution

forbid us to combine ^{35}S Met with cell cultures (questions of amount of radioactivity, combining safety levels and waste management).

Figure R2. Does WAVE2 escape from the WRC ? The protocol combines electroporation of HeLa cells with purified PC-tagged BRK1 homotrimers to monitor assembly of the WRC around the exogenous BRK1 with pulse-chase after metabolic labeling of neo-synthesized subunits. The Coomassie stained gel indicates that newly assembled complexes represents a minority of all WRCs. However the autoradiography indicates that the newly assembled complexes combines most of neo-synthesized subunits, at least for CYFIP1, NCKAP1 and ABI1. The majority of neosynthesized WAVE2 is found associated with the bulk of WRC, as if WAVE2 was able to exchange between WRCs.

During the revision, we thus attempted to monitor WAVE exchange between WRCs, by using a transmembrane subunit of the WRC (CYFIP1) expressed by mouse 3T3 cells (in a manner similar to the strategy used in Mehidi et al., Nat Cell Biol 2021). We wanted to take advantage of the availability of custom-made mouse- and human-specific WAVE2 antibodies to detect the association of human WAVE2 coming from a MCF10A extract with the transmembrane WRC derived from the mouse cells. To immobilize transmembrane WRC, we have permeabilized adherent 3T3 cells and incubated these “ghosts” with MCF10A cytosolic extracts in the presence of an energy-regenerating mix. Unfortunately, this assay did not allow us to detect the WAVE2 exchange (we spare to the reviewer all the cumbersome attempts that were made). So at this point, we believe that WAVE2 exchanges between WRCs in living cells, but that this exchange event is not yet reconstituted in an in vitro assay. To dissect this ‘transparent’ phenomenon of WAVE exchange is one of our future directions, since we believe like the reviewer that the WSC is likely to play a role in this phenomenon and that it is probably important for WAVE activation. We are now convinced that to reveal subunit exchanges between different complexes in the cell will require long methodological developments and that it is beyond reasonable revision of the present manuscript.

7. The bead assay with Rac-L61 is not very conclusive as it suggests a huge role for PPP2R1A which is not reflected by the other assays. According to the bead assay PPP2R1A would be almost essential for WRC mediated actin assembly although all other assays show that it is just a modulator. Is it possible to test which proteins are recruited to the beads during that assays? PPP2R1A RNA interference has a dramatic effect despite the presence of constitutively active L61 Rac on the beads, which abrogates/redundantizes the effects of PPP2R1A according to the migration assays and Abi pull downs. How can this be reconciled?

We agree with the reviewer that in the original version, the bead assay appeared non consistent with migration results, since there was no effect of PPP2R1A depletion on the persistent migration of RAC1 Q61L expressing cells and that there was a strong effect of PPP2R1A depletion on actin polymerization triggered at the surface of the beads displaying RAC1 Q61L. We have figured the reason of the discrepancy during the revision. The branched actin structures induced at the surface of beads do not directly depend on immobilized RAC1 Q61L, but rather on endogenous RAC1 ! We were able to confirm this important observation using two independent approaches, by treating cell extract with the RAC1 inhibitor NSC23766 and by depleting them of the endogenous RAC1 using siRNA-mediated depletion. These data have been incorporated in the new Fig.7.

The important implication is that results from the in vitro assay are now compatible with cell migration results, where PPP2R1A is required when the feedback loop is required. Furthermore, the bead assays now provide a rather direct evidence of the RAC1 positive feedback loop. The fact that RAC1 Q61L requires endogenous RAC1 means that RAC1 Q61L activates endogenous RAC1, and therefore that RAC1 activates itself in positive feedback. We warmly thank the reviewer for this comment that allowed us to figure this out. We believe that this experimental result is a major addition to our manuscript. The arguments in favor of the feedback loop are now recapitulated in the last paragraph of the discussion.

We also used stable MCF10A cells expressing GFP-ABI1, GFP-PPP2R1A or GFP-NHSL1 (obtained especially for this purpose) to prepare cell extracts to incubate in the bead assay. We observed a strong recruitment of ABI1 and NHSL1 in agreement with the GST pull-down assay and no enrichment of PPP2R1A in line with the limited association with RAC1 Q61L beads. These data were incorporated in the Supplementary Figure 7 (Fig.S7c).

8. The sentence in lines 348-50 is also not helpful in this respect. (“The in vitro reconstitution assay of RAC1- dependent branched actin structures we report here using cell extracts is likely to play an important role in testing different scenarios for a multistep WAVE activation cycle”) Why bother to show an inconclusive assay here when it is only becomes important much later when we know more?

The sentence has been removed. We hope that the reviewer finds the bead assay more conclusive now (point #7, right above). But this is true that there is no need to announce what might be important in the future.

9. Phosphorylation studies: It was a brave of the authors to attempt finding changes in the phospho-proteome of lamellipodial components. However, it is not so surprising that they did not find any. As the authors point out correctly, PPP2R1A might either play a scaffolding role without its catalytic subunit or the authors simply did not find the important (de) phosphorylated residue. The answer might even be Rac itself, which can be inhibited by phosphorylation on Ser71 (Kwon et al., 2000). This modification can be probed by Western blotting.

Dephosphorylation via PP2A would explain the Rac activation-effect described earlier (and see above). The text part on WAVE Ser308 should be toned down or removed as this is a very unlikely candidate (PPP2R1A interacts with WSC not WRC).

We agree with the reviewer's suggestion. We have moved the phosphoMS analysis to the Supplementary Figure 6. We have enriched this figure with analyses of RAC1 phosphorylation, which was also requested by another reviewer (see also point #12 of reviewer #1). There was no difference on phosphorylation of RAC1 serine 71 when PPP2R1A was knocked down, knocked out or overexpressed.

We also found that PP2A phosphatase inhibitors were not having any effect in the *in vitro* actin polymerization assay. These new data were added to the main figure 7 (Fig.7c). The apparent lack of role of PP2A phosphatase activity is now discussed in the first paragraph of the discussion.

10. The localization studies: The images in Fig 3 actually show that there is no co-localization between PPP2R1A and NHSL1. Whereas NHSL1 clearly is a tip component PPP2R1A is not. It even does not seem to fully recapitulate the actin-Arp2/3 complex-gradient. The authors need to show by e.g. FRAP assays, if the turnover of the two proteins at the lam tip speaks for interaction and that on the other hand the portion of PPP2R1A further back in the lamellipod is turned over at other rates than PPP2R1A at the lam tip. Is PPP2R1A turnover similar to that of NHSL1 in the places of putative colocalization?

We followed the suggestion of the reviewer and performed FRAP of PPP2R1A in B16F1 cells. GFP-PPP2R1A recovers fast and homogeneously throughout the lamellipodium with a $t_{1/2}$ on the order of 1 s. There is no difference in the recovery time at the lamellipodium tip and in the width of the lamellipodium. These results are consistent with the idea that PPP2R1A is simply associated with actin networks of the lamellipodium, but not a structural component of branched actin networks.

We also performed simultaneous FRAP of GFP-NHSL1 and mScarlet-PPP2R1A. The $t_{1/2}$ of both proteins at the lamellipodium tip are similar. This is compatible with the idea that the two proteins detected at this location are indeed forming a complex.

Both FRAP analyses are displayed in a new main figure, Fig.5.

We thank the reviewer for his or her careful analysis of our results and constructive criticism. The main improvements are the following:

- Measurements of RAC1 levels and activity
- The bead assay requires the activity of endogenous RAC1, but not of PP2A
- Relative abundance of WSC/WRC and distribution of complexes in sucrose gradients
- FRAP analyses of PPP2R1A and NHSL1

In summary, the results obtained during the revision generated 2.5 new main figures (Fig.3 reconstitution, Fig.5 FRAP, Fig.7 bead assay), 4 new supplementary figures (Fig.S1 RAC1 assays, Fig.S3 coiled coil, Fig.S4 WSC levels, Fig.S5 sucrose gradients) and two new movies (S7 and S8 for FRAP).

REVIEWERS' COMMENTS

Reviewer #1 (Remarks to the Author):

The authors have addressed my previous questions, and I do not have any additional comments.

Reviewer #2 (Remarks to the Author):

Although the role of the NHL1 complex that contains the CYFP/ABI1/BRK1/NAP assembly is not clearly defined in cell migration, the authors did an excellent job clarifying several issues thereby making the manuscript conclusions more cogent. It will be interesting to see a follow-up in this direction.

The interpretation of the role of RACQ61L on directional persistence is still uncertain and somewhat confusing. Some rewording of the text on these aspects appears necessary

Reviewer #3 (Remarks to the Author):

After reading the revised manuscript and the point by point reply, I am now satisfied and convinced that this manuscript is an important contribution to our understanding of WRC together WSC and their regulation. I was particularly pleased to see that the authors attempted to answer every single comment of all three reviewers experimentally and also that they were willing to reword and sharpen their conclusions not only along the recommendations but - more importantly - based on the new experimental data. I particularly like the way the Rac-bead assay turned out which was an 'outliner' in the earlier version.

Also I agree with the authors that it is the nature of science to never be able to answer all questions. Nonetheless, all new data substantiated the findings and strengthened the conclusions. For the reviewer this is now a conclusive and important piece of work, which can be recommended for publication.